# Targeting HSP90 as a Novel Therapy for Cancer: Mechanistic Insights and Translational Relevance

**DOI:** 10.3390/cells11182778

**Published:** 2022-09-06

**Authors:** Jian Zhang, Houde Li, Yu Liu, Kejia Zhao, Shiyou Wei, Eric T. Sugarman, Lunxu Liu, Gao Zhang

**Affiliations:** 1Institute of Thoracic Oncology and Department of Thoracic Surgery, West China Hospital of Sichuan University, Chengdu 610041, China; 2Western China Collaborative Innovation Center for Early Diagnosis and Multidisciplinary Therapy of Lung Cancer, Sichuan University, Chengdu 610041, China; 3Faculty of Dentistry, The University of Hong Kong, Prince Philip Dental Hospital, 34 Hospital Road, Sai Ying Pun, Hong Kong 999077, China; 4Philadelphia College of Osteopathic Medicine, Philadelphia, PA 19131, USA

**Keywords:** heat shock protein 90, INHIBITORS, cancer therapeutics, translational relevance

## Abstract

Heat shock protein (HSP90), a highly conserved molecular chaperon, is indispensable for the maturation of newly synthesized poly-peptides and provides a shelter for the turnover of misfolded or denatured proteins. In cancers, the client proteins of HSP90 extend to the entire process of oncogenesis that are associated with all hallmarks of cancer. Accumulating evidence has demonstrated that the client proteins are guided for proteasomal degradation when their complexes with HSP90 are disrupted. Accordingly, HSP90 and its co-chaperones have emerged as viable targets for the development of cancer therapeutics. Consequently, a number of natural products and their analogs targeting HSP90 have been identified. They have shown a strong inhibitory effect on various cancer types through different mechanisms. The inhibitors act by directly binding to either HSP90 or its co-chaperones/client proteins. Several HSP90 inhibitors—such as geldanamycin and its derivatives, gamitrinib and shepherdin—are under clinical evaluation with promising results. Here, we review the subcellular localization of HSP90, its corresponding mechanism of action in the malignant phenotypes, and the recent progress on the development of HSP90 inhibitors. Hopefully, this comprehensive review will shed light on the translational potential of HSP90 inhibitors as novel cancer therapeutics.

## 1. Introduction

The efficient and accurate control of the cellular protein pool is crucial for homeostasis within the crowded environment of a single cell [1]. HSP90 is one of the heat shock protein members. When functionally upregulated under environmental stress, HSP90 protects cells from detrimental effects [1]. HSP90 is evolutionarily conserved and ubiquitously expressed across species, which accounts for 1–2% of total cellular proteins in the unstressed condition [2]. It can be further increased to about 4–6% in the stressed condition [3]. Due to its abundance and adhesive properties, HSP90 has been likened to “molecular glue” [4]. The structure of HSP90 comprises three domains: the N-terminal domain (NTD) with ATPase activity, the middle domain (MD) that binds to the client protein, and the primary dimerization C-terminal domain (CTD) [1,5]. HSP90 utilizes ATP to keep its “closed” conformation for the binding of client proteins. The immature client proteins proceed to fold, accompanied by ATP hydrolyzation and energy release. After that, HSP90 releases the matured product and is transformed into an “open” conformation. Co-chaperones are also required for the intricate control of ATP hydrolysis rates and the certain conformational states [1,6,7].

Over 20 co-chaperones of HSP90 have been documented in eukaryotic cells, which modulate the molecular functions of HSP90 in four major ways: (1) coordinate the interplay between HSP90 and other chaperone systems, such as HSP70; (2) stimulate or inhibit the ATPase activity of HSP90, i.e., AHA1 for stimulation and CDC37 for inhibition; (3) recruit specific classes of clients to HSP90; (4) contribute to various aspects of the chaperone cycle through their enzymatic activities (Figure 1). The co-chaperones containing the TPR domain (i.e., HOP) facilitate the cooperative and successive action of the HSP90-HSP70-HSP40 complex to achieve the maturation of client proteins. Separately, the co-chaperones that inhibit the ATPase activity are more likely to be involved in client loading or the formation of mature HSP90 complexes, whereas those that enhance the ATPase activity are regarded as activators of the HSP90 conformational cycle [1].

Recently, Taipale M. et al., systematically screened 2156 clones, including protein kinases, transcription factors, and E3 ligases, to identify HSP90’s dependency. They found that more than 400 client proteins depend on the protein folding machinery regulated by Hsp90 for the achievement and maintenance of their active conformations [10]. Additionally, a number of studies demonstrated that the HSP90 client proteins also regulate various cellular functions, including signal transduction, protein trafficking, chromatin remodeling, autophagy, and cell proliferation and survival [11,12,13,14,15,16,17,18,19]. Furthermore, many HSP90 client proteins are frequently mutated and/or over-expressed in cancer cells. Therefore, they have been actively pursued as therapeutic targets for cancer treatment [12].

## 2. Subcellular Localization of HSP90

HSP90 is a highly conserved molecular chaperone on the evolutionary level and is constitutively expressed in most organs and tissues [20]. HSP90 assists in the proper folding, intracellular disposition and proteolytic turnover of many key regulators of cellular homeostasis with its ATPase activity [12,21,22,23]. HSP90 is mainly located in cytoplasm under normal conditions, where polypeptides are synthesized, and aberrantly folded proteins are produced. By using an in vivo PET tracer or in vitro organelle fractionation assay, HSP90 was found to be expressed ‘everywhere’, including the nucleus, mitochondrion, and plasma membrane, as well as also being secreted into the extracellular matrix [24,25,26]. The differential expression of sub-cellular HSP90 may exert distinctive functions in multiple biological processes, especially tumorigenesis [25]. Interestingly, the differential expression pattern of HSP90 between normal cells and cancer cells has already been documented, with the discrepancies selectively existing in the mitochondria or extracellular matrix [26,27].

### 2.1. Cytosolic Localization of HSP90 

HSP90, as a critical homodimer chaperone machinery, was expressed within the cytosol component [28]. By constructing various truncated forms of HSP90 and using a confocal microscopy, Passinen S. et al., found that the C-terminal half of HSP90 (amino acids between 333 and 664) is responsible for the cytoplasmic localization [22]. Nonetheless, by fusing with the nuclear localization signal sequence (NLS), NLS-HSP90 was preferentially expressed in the nucleus [22,28].

Previous studies have focused on the discrepancy of HSP90 expression between normal cells and cancer cells and demonstrated that a higher protein level of HSP90 is expressed in the latter. In these studies, the majority of HSP90 was also localized in the cytoplasm [23,27]. Perhaps the cytosolic HSP90 largely contributes to the folding/refolding of critical oncogenic drivers and pro-survival regulators by supplying a co-chaperone buffering system. Additionally, cytosolic HSP90 has also been termed “molecular glue”, highlighting its abundance and buffering properties [4]. Most of the client proteins of HSP90 are located in the cytoplasm, exerting a variety of functions, including signaling transduction, post-transcriptional modification, metabolic rewiring and cytoskeleton remodeling. Recently, HSP90 was found to transiently crosslink actin filaments in vitro, and this dynamic interaction between HSP90 and almost all cytoplasmic filamentous structures highlights its role in modulating actin filament bundling behavior [4,29]. HSP90 was also associated with tubulin and probably protects it from heat denaturation [30]. Likewise, it was shown that HSP90 protects myosin from heat stress [31]. Therefore, HSP90 is functionally involved in the management of the cytoskeleton.

### 2.2. Nuclear Localization of HSP90

A low expression of HSP90 (5–10% of total cellular HSP90) was detected in the nucleus of normal cells [32]. Recent data, however, have shown that the elevated expression of nuclear HSP90 could be detected in breast cancer and non-small cell lung cancer (NSCLC) [33,34,35]. For example, a nuclear accumulation of heat shock protein 90 might predict the poor survival of patients with NSCLC. Furthermore, the nuclear staining of HSP90 was also positively correlated with the age and smoking status of patients with NSCLC [34]. Interestingly, in quiescent *Saccharomyces cerevisiae* cells, Hsp90 and its co-chaperones were found to accumulate in the nucleus with the requirement of the α/β importin system, which was enhanced during periods of relative metabolic inactivity [36]. In fact, the HSP90 protein comprises sequences that are homologous to the recognized traditional or alternative nuclear import and export signals: the nuclear localization sequence [21].

Importantly, HSP90 is associated with multiple nuclear chaperone clients including nucleic acid, histone, transcription factors, and epigenetic regulators [4]. HSP90 also modulates multiple biological functions in the nucleus including RNA synthesis, processing, and multiple telomerase activities [4,37]. The nuclear translocation of HSP90 is governed by FKBP52, steroid receptors, and kinases [21,38]. Zinc finger proteins, helix-loop-helix proteins, MyoD1, E12, HIF1α, HSF1, and glucocorticoid receptor interact with HSP90 [4,39,40]. The transfection of 3T3 cells with HSP90 fused with EGFP revealed that HSP90 was located in the nuclear membrane upon exposure to an elevated temperature [41]. Many small chemical molecules enter the cytoplasm through penetrating the plasma membrane, but it was shown that none of them could enter the nucleus, rendering their efficacy limited [42]. However, HSP90 could be translocated from cytoplasm to nucleus, which protects cancer cells from therapeutic pressure [43]. This suggests that nuclear-directed HSP90 inhibitors should be taken into consideration.

### 2.3. Mitochondrial Localization of HSP90 

The mitochondrial expression of HSP90 was unexplored until Kang B. et al., first demonstrated the high expression of HSP90 in tumoral mitochondrion, but barely in normal tissues [26]. Based on this ground-breaking work, substantial research took place, which focused on the development of small molecular drugs with the organelle-specific targeting of HSP90. Collectively, the goal of these inhibitors is to trigger a sudden collapse of mitochondrial integrity and apoptosis that would selectively occur in tumor cells. Several inhibitors that specifically target mitochondrion have emerged and shown intriguing effects in multiple cancer types, including pancreatic cancer, breast cancer, colon cancer, NSCLC, melanoma, glioblastoma, prostate cancer, lymphoma, and leukemia (Table 1) [26,44,45,46,47,48,49,50,51]. By investigating the metabolic network in the tumoral mitochondrion regulated by HSP90, Chae Y. et al., highlighted that mitochondrial HSP90 (hereafter mtHSP90), but not cytosolic HSP90, binds and stabilizes the electron transport chain complex II subunit succinate dehydrogenase-B (SDHB), which maintains cellular respiration under low-nutrient conditions and contributes to HIF1α-mediated tumorigenesis in patients carrying SDHB mutations [52]. Cryo-EM data also confirmed the dynamic interplay between mitochondrial HSP90 and SDHB folding intermediates [53].

It is worth mentioning that TRAP1 has been regarded as another version of HSP90—namely, mitochondrial HSP90—because it shares 60% of its sequence with HSP90 and contains the same domains: NTD, ND, and CTD [32,95]. Interestingly, Kang B. et al., also established that TRAP1 was consistently elevated in primary tumors, whereas it was nearly undetectable in normal tissues [26]. Moreover, one of the important client proteins of TRAP1 was cyclophilin D, a mitochondrial residential protein, which maintains mitochondrial integrity by preserving cells from apoptosis. Furthermore, the crystal structures of HSP90 and TRAP1 provide further molecular insights that can be exploited for the development of novel inhibitors [96,97]. Taken together, TARP1 is another attractive target for developing cancer drugs [98].

### 2.4. Membrane and Extracellular Localization of HSP90

HSP90 was first viewed as an artifact when it was found on the cell surface by a functional screening, due to its abundant expression within the cells. After cautious verification, HSP90 is no longer considered as being exclusively located within the cell [75]. HSP90 can also be secreted into the extracellular matrix. Accordingly, the cell surface expression of HSP90 is higher on cancer cells than that on normal cells, which correlates with the malignant stage of the tumor. Previous work has also shown that the cell surface of HSP90 could strengthen the migration potential of cancer cells that is distinct from the function of the intracellular HSP90 pool [99,100]. Thus, the cell surface of HSP90 is also an attractive therapeutic target in terms of inhibiting tumor invasion and metastasis [75,101].

The first evidence for the detection of HSP90 in the extracellular matrix was implicated in 1986 when Barrott J. et al., found a mouse tumor-specific antigen that was identified as a heat shock protein, which is now recognized as HSP90 [102]. Since the ATP level in the extracellular environment is low due to the lack of energy source, the extracellular HSP90 (hereafter eHSP90) may function independently of ATP. The secretion of eHSP90 is induced by environmental stresses and growth factors [64,103], and it is affected by post-translational modifications to the chaperone, including phosphorylation and acetylation [104]. Recent work has also shown that HSP90α is released by invasive cancer cells via exosomes, which contributes to their invasive nature by interacting with plasmin [69]. Another study found that both HSP90α and HSP90β are secreted by cancer cells to interact with MMP2 and MMP9 to enhance the invasive capacity of tumor cells. Similarly, eHSP90 was detected in normal cells only in response to stress, while cancer cells consecutively secrete HSP90 [105,106]. Interestingly, eHSP90 interacts with a series of receptors such as EGFR/HER2/LPR1 to promote the downstream signal transduction associated with tumor growth and metastasis, which resembles the EMT phenotype [107,108,109]. Further, eHSP90 expression correlates with an increase in metastatic potential and a decrease in the immune response in multiple cancer types [100]. Although the specific inhibition of eHSP90 does not affect cancer cell growth in vitro or tumour xenograft progression in vivo [99], the inhibition of eHSP90 is effective in conquering metastasis with minor side effects, highlighting the clinical potential of eHSP90 inhibitors [49,101,106,110,111].

## 3. HSP90 and Its Clients in Cancer Phenotype

HSP90 is a hub in the network of molecular chaperone cycles because it promotes both the folding and degradation of various client proteins, in addition to regulating the expression of other quality control components (Table 1). HSP90 plays a significant role because it is involved in regulating signal transduction, protein trafficking, receptor maturation, and innate and adaptive immunity, as well as nearly all the hallmarks of cancer. Those complex processes can be achieved successfully by the interplay of HSP90 with its co-chaperones [112].

### 3.1. Uncontrolled Proliferation

Sustained proliferative signaling is one of the most important hallmarks of cancer. While multiple regulators of cell growth signaling are the clients of HSP90, it is the HSP90 chaperone complex that maintains the signaling circuitry critical for the independent growth of tumor cells.

HSP90 and its ER homolog, GRP94, bind to and stabilize HER2. The treatment of benzoquinone ansamycins (BA, a HSP90 inhibitor) in COS7 cells disrupts this association, thus, resulting in a rapid poly-ubiquitination of HER2 followed by the proteasome-dependent degradation of HER2 [113]. Further, cytosolic Akt is associated with HSP90 and CDC37, which form a complex. A functional study also showed that HSP90 is required for Akt stability. The treatment of ATP-binding inhibitors in MCF-7 and SKBr-3 cells leads to the ubiquitination and degradation of Akt [73]. Similarly, the translocation of BA to HSP90 disrupts its association with CDK4, resulting in a reduction in the half-life of newly synthesized CDK4 [55]. A subsequent study performed in insect cells also showed that CDC37 drives HSP90 to target CDK4 both in vitro and in vivo, and HSP90 preferentially binds to CDK4, which is not associated with D-type cyclins. Moreover, disruption of the CDC37-HSP90 complex and its function by specific inhibitors results in an unstable state of CDK4, indicative of the role of HSP90 in protecting its client proteins from proteasome degradation [55]. RAF1 kinase was also associated with HSP90, and it has been shown to stabilize RAF1 and prevent it from 26S proteasome-mediated degradation. The treatment of HSP90 inhibitor in NIH 3T3 cells impairs the RAF1-MEK signaling, both by disabling RAF1 to reach the plasma membrane and disrupting the RAF1/MEK1 interaction [54]. The onco-fusion protein BCR-ABL also associates with HSP90. Thus, the treatment of BA inhibitor in Bcr-Abl-expressing HL60 cells induces the degradation of BCR-ABL, which may re-sensitize leukemia cells expressing the BCR-ABL to chemotherapy [56,114]. Collectively, HSP90 regulates the proper folding of mutated or overexpressed protein kinases and protects them from degradation by blocking their association with poly-ubiquitination mediated 26S proteasome [23].

### 3.2. Anti-Apoptosis and Immortalization

p53 is a critical checkpoint to trigger growth arrest and apoptosis when numerous genetic disorders or DNA damage is induced by ultraviolet radiation or chemotherapeutics. Nevertheless, the inactivating and mutated form of p53 appears to be dominant-negative by imposing a defective heterodimer conformation with wild-type p53. HSP90 is increased in tumors to chaperone with mutated p53 and to stabilize the altered conformation of mutant p53, thus, protecting it from proteasome degradation [115,116]. Blagosklonny M. et al., demonstrated that the treatment of several cancer cell lines with BA resulted in the destabilization of mutated p53 with no influence on wild-type p53 forms, thus, leading to the restoration of the activity of the wild-type p53 in p53 heterozygous tumor cells [117].

HSP90 was also shown to associate with and stabilize PKM2 in hepatocellular carcinoma cells. HSP90 interacts with GSK-3β and promotes its activity, thus, increasing the phosphorylation of PKM2 at Thr328. This process was critical for maintaining the stability and biological functions of PKM2 [118].

BCL-2 was the first apoptotic regulator identified in organism, which localized to the outer membrane of mitochondria. It plays an important role in promoting cellular survival and antagonizing the apoptotic complex. HSP90 promotes the survival of leukemia cells by binding to APAF-1 and BCL-2. Blockage with HSP90-specific inhibitor geldanamycin (GA) also diminishes the association of HSP90 with APAF-1 or BCL-2 in leukemia cells, leading to apoptosis [119].

It has been identified that the molecular chaperone HSP90 binds to the catalytic subunit of telomerase, and this interaction enhances the assembly of active telomerase. Using an in vitro assay, researchers also found an abundance of active telomerase from cell extracts, which was associated with HSP90 [18]. Consistent with these in vitro results, HSP90 facilitates DNA extension by promoting telomerase assembly and occupancy both in budding yeast and human cells [37,120].

### 3.3. Invasion, Metastasis, and Angiogenic Role

As a modality of malignant behavior, the invasion or migration of tumor cells occurs even in the early stage of tumor progression. This biological process is complex and often accompanied by the remodeling of a number of proteins. We discuss in the “Membrane and Extracellular HSP90” section that the secreted HSP90 interacts with MMP2 and MMP9 to promote invasive phenotype [105]. Furthermore, the treatment of MDA-MB-453 breast cancer cells with monoclonal antibodies against HSP90 significantly inhibits the metastatic potential by disrupting the interaction of MMPs with HSP90 [106]. The depletion of HOP by the RNA interference approach, which is a co-chaperone protein that binds to both HSP70/HSP90, also reduced the invasiveness of pancreatic cancer cells by decreasing the expression of matrix metalloproteinases-2 [121]. Epithelial to mesenchymal transition (EMT) was strongly associated with cancer invasiveness and metastasis. Treatment with a HSP90 inhibitor, ganetespib, or the knock-down of *HSP90* down-regulated genes associated with EMT, invasion, and motility, indicate a strong relevance of HSP90 with an aggressive EMT phenotype [122]. MET was also shown as a client protein of HSP90 chaperone, which enhances the pro-invasive role of HSP90 by interacting with HSP90 [123]. Taken together, HSP90 is important for the highly invasive and metastatic potential of tumor cells by chaperoning multiple key factors such as receptor tyrosine kinases and EMT-related transcription factors [19].

It is well known that tumor growth requires both nutrients and oxygen. Tumor cells produce a high level of HIF1α, which can mediate the expression of VEGF and other pro-survival factors. However, HIF1α was associated with the molecular chaperone HSP90. NVP-AUY922, a novel HSP90 inhibitor, promotes HIF1α degradation via VHL E3 ubiquitin ligase [124]. Another HSP90 inhibitor, AT-533, also inhibits HIF-1α/VEGF and its downstream signaling pathway in breast cancer cells [76].

### 3.4. Others

Glucocorticoid receptor (GR) was reported to associate with HSP90 by its hormone binding domain aligned with amino acids from 568 to 616 [77]. This was regulated by the deacetylation of HSP90 by HDAC6, a reversible post-translational process critical for the maturation of GR [78]. Similarly, steroid hormone receptors, mineralocorticoid receptors, progesterone receptors, estrogen receptors, androgen receptors, and oestrogen receptors form a complex with HSP90 to protect its chaperone role, which in turn, promotes their maturation and the normal function of signaling cascades. The structure and function of those receptors are also affected by the occupancy of specific HSP90 inhibitors [79,80,81,82,83]. Cystic fibrosis transmembrane conductance regulator (CFTR) was associated with HSP90, and the treatment of geldanamycin in BHK cells expressing wild-type CFTR accelerated the degradation of CFTR [84]. It has also been shown that HSP90 is associated with endothelial nitric oxide synthase (eNOS) and enhances its activation [85]. The stability of epigenetic regulators such as UHRF1 and BRCA2 were regulated by HSP90 machinery [86,87]. HSP90 was implicated with cooperating with transcriptional factors such as Nanog, Oct4, JAKs, and Stat3 to control the stem cell pluripotency as well as the activation of cytokine signaling [88,89]. The requirement of HSP90 to associate with calcineurin and Tau proteins in the pathological condition makes it an attractive target in glioblastoma and neurodegenerative diseases [90,91]. In plants, HSP90 is also conserved and essential for regulating NLR (nucleotide-binding domain and leucine-rich repeat containing) proteins [92]. Unlike its traditional role of protection, HSP90 also prevents MDM2 from interacting with mutant p53 in order to promote the degradation of oxidized calmodulin, which demonstrates the versatility of HSP90 [93,94,125].

## 4. HSP90 Inhibitors in Cancer Therapeutics

HSP90 serves as a molecular chaperone for a variety of client proteins, including receptor tyrosine kinases, metabolic enzymes, and epigenetic regulators that are critical for the proliferation and survival of cancer cells. Most client proteins with oncogenic roles are over-expressed in various types of cancer cells. Therefore, the development of small molecules that specifically target the HSP90 itself or disrupt the association of HSP90 with its partner oncoproteins is urgently needed. Indeed, numerous HSP90 inhibitors have been identified or developed in the last two decades with different mechanisms of action. Here, we review the chemical structure, binding site, organelle selectivity, and application of currently developed HSP90 inhibitors and the clinical setting in Table 2. Generally speaking, there are two categories of these inhibitors: (1) direct HSP90 inhibitors; (2) HSP90/co-chaperone inhibitors.

### 4.1. HSP90 Inhibitors That Directly Bind to HSP90

#### 4.1.1. Geldanamycin (GA) and Its Derivatives

Geldanamycin, a member of the ansamycin class of antibiotics, was originally isolated from the bacterium *Streptomyces hygroscopicus*, and turned out to be the first HSP90 inhibitor resulting from natural product-based drug discovery [182]. Geldanamycin showed a possible anti-tumor effect on several cancer cell lines. The molecular target, however, was unknown until Whitesell L. et al., found that geldanamycin could target HSP90 by mediating the degradation of v-Src [183]. Unfortunately, the subsequent pre-clinical studies revealed hepatotoxicity, low chemical stability, poor bioavailability, and the solubility of geldanamycin, which largely limited its further translational advancement [184]. Fortunately, researchers improved the unfavorable properties of geldanamycin by developing a series of its derivatives [185]. Among those modified GA derivatives, 17-AAG (17-allylamino-GDA) and 17-DMAG (17-(2-Dimethylaminoethyl)amino-17-demethoxygeldanamycin) emerged, owing to their excellent inhibitory potency and solubility. Therefore, both of the GA derivatives entered into clinical trials for further evaluation [128,186,187]. The future of GA research should focus on addressing the toxicity of benzoquinone as well as identifying the combination therapeutics.

To test the impact of network subcellular compartmentalization on the activity of HSP90 inhibitors, Kang B. et al., designed another analog of 17-AAG named gamitrinib-TPP (GA mitochondrial matrix inhibitor). Gamitrinib-TPP is combinatorial and contains a benzoquinone ansamycin backbone derived from 17-AAG, a linker region on the C17 position, and a mitochondrial targeting moiety. Gamitrinib was accumulated in the mitochondria, which caused a rapid tumor regression due to its “mitochondriotoxic” mechanism of action [138]. Subsequent studies also showed the substantial anti-tumor activity of gamitrinib both in vitro and in vivo while sparing the normal counterpart [139,188,189,190,191].

#### 4.1.2. Radicicol and Its Derivatives

Radicicol, a resorcinol lactone antibiotic, was first found in *Monosporium bonorden* by P. Delmotte and J. Delmotte-Plaquee in 1953 [192]. Later on, radicicol was found to manifest anti-tumor properties, possibly by inhibiting v-Src and its downstream MAPK signaling [193]. Radicicol showed a better affinity for HSP90 versus geldanamycin in vitro. Nevertheless, further translational research on radicicol has been limited because radicicol was rapidly metabolized to an inactive form due to its electrophilic nature [155,194]. Hence, radicicol was not suitable for being tested in clinical trials. However, the replacement of epoxide by a difluorocyclopropane ring generated lesser anti-tumor activity compared with radicicol. The replacement of 2′-ketone by an oxime produced KF25706, which was metabolically stable and exhibited in vivo anti-tumor activity [195]. It was found that the oxime derivatives also contain a pharmacophore which could be used for designing a series of radicicol analogs [196,197].

Molecular studies also demonstrated three dynamic conformations of radicicol: the bioactive “c-shaped” conformation, the planar conformation, and the conformation where the macrocycle is bent to the opposite of the resorcinol ring [198]. Ganetespib (STA-9090) showed greater tumor penetration and more mild side effects than that of 17-AAG, and it is now under clinical evaluation in phase I-III trials [199,200,201].

Interestingly, Cheung K. et al., conducted a high-throughput compound screening and found a resorcinol-containing compound, CCT018159, which manifested the inhibitory activity of HSP90 [150,202]. A further medicinal chemistry modification based on CCT018159 led to the development of the AUY922, which is currently under evaluation in a Phase II trial [203,204,205]. KW-2478, a resorcinol-derived molecule, which was developed based upon a unique optimization strategy, also entered clinical investigation for patients with B-cell malignancies and refractory multiple myeloma [152,206]. Similarly, Astex Pharmaceuticals developed AT13387 based on a fragment-based approach, which was tested in metastatic solid tumors due to its anti-HSP90 property [153,207].

#### 4.1.3. Chimeric Molecules

Both geldanamycin and radicicol confer substantial inhibitory effects on various types of cancer cells, but they are of limited translational potential due to the toxicities of GA and the insufficient in vivo efficacy. Therefore, researchers have proposed the development of new HSP90 inhibitory scaffolds by combining the structural features found in both geldanamycin and radicicol [184]. This approach would ensure the preservation of the moiety interacting with HSP90, while simplifying the structure. Radanamycin was the first chimeric HSP90 inhibitor which retained the hydrogen bonding network responsible for the selective binding of the heteroprotein complex [196]. Radamide, a radanamycin analog, was synthesized by connecting the resorcinol ring to the quinone via an amide bond containing carbon-linker [208]. Radester, another analog, was produced when the radicicol ester was connected to quinone [209]. All three compounds showed a greater HSP90 binding affinity and enhanced anti-proliferative activity.

#### 4.1.4. Purine-Based Molecules

The functional HSP90 requires the addition of ATP. To capitalize on this, Chiosis G. et al., utilized ATP as a starting point to design small molecule inhibitors that bind to the N-terminal ATP-binding site [210]. This led to the development of the first purine-based inhibitor, PU3, which also manifested a comparable anti-tumor activity by antagonizing HSP90 [210,211]. Consequently, multiple HSP90 inhibitors were developed but only the substitutions at the 2-, 4-, 5- and 9-positions are critical for inhibiting HSP90 [212]. Among all analogs, BIIB021 represented one of the HSP90 inhibitors with the most efficacious activity and showed a high selectivity for tumor versus normal cells in the ERBB2 degradation assays with a pyridylmethylene group at the 9-position [157]. Further, adding a phosphate ester led to the development of BIIB028, and both of the two analogs entered clinical trials [159,213].

#### 4.1.5. FDA-Approved Inhibitors Target HSP90

Though inhibitors that directly bind to HSP90 are still under evaluation in clinic trials, researchers found that two FDA-approved inhibitors (panobinostat and irsogladine) exhibit an HSP90 inhibitory effect [214,215]. Panobinostat, developed by Novartis, has primarily been recognized as a histone deacetylase (HDAC) inhibitor. Oral panobinostat is approved in the US, as combination therapy with bortezomib and dexamethasone in patients with recurrent multiple myeloma who have received at least two prior treatment regimens [216]. However, panobinostat induces the hyperacetylation of HSP90 in acute myeloid leukemia cells and inhibits its chaperone function, thereby leading to the proteasomal degradation of client proteins such as CXCR4 and AML1/ETO9a [217,218]. Moreover, the treatment of multiple myeloma with panobinostat induced the degradation of PPP3CA, which inhibited the cell viability [219]. Irsogladine, first named as MN-1695, showed remarkable efficacy in various animal models of gastric ulcers [220]. Irsogladine is a phosphodiesterase inhibitor approved by the FDA for its mucosal protective efficacy in the treatment of peptic ulcers and acute gastritis [221]. Interestingly, Young Ho Seo screened FDA-approved drugs based on the similarity of chemical structures and selected drugs that contained both a hydrophilic and a hydrophobic binding moiety, discovering that irsogladine acts as an HSP90 inhibitor and disrupts the HSP90 folding machinery [215]. These studies have laid a solid foundation for the further development of potential inhibitors that may bind to HSP90.

#### 4.1.6. Other Inhibitors

Epigallocatechin-3-Gallate (EGCG) was a polyphenolic compound that was originally isolated from green tea and showed affinity for the C-terminal ATP-binding site of HSP90 at residues 538–738 [222]. The subsequent studies demonstrated that the phenols on the B- or the D-rings are unfavorable, while the syn-stereochemistry of the linker that connects the same rings with the benzopyran core is indispensable for its HSP90 inhibitory activity [223]. Silybin was first isolated from the seed of *Silybum marianum.* It showed anti-tumor activity by inhibiting the growth of cancer cell lines, possibly by promoting the degradation of multiple HSP90-dependent client proteins, including HER2, RAF1, Akt, CDK2, CDK4 [224,225]. Cisplatin was a classical chemodrug and widely used for the treatment of a number of solid tumors because it could be inserted into double-stranded DNA to form adducts, which produced a lethal effect [166,226]. Interestingly, the following studies demonstrated that the cisplatin could bind to the C-terminal domain of HSP90 and inhibit its chaperone activity [227,228]. Recently, an optimized derivative of cisplatin, LA-12, was developed, which exhibited a greater binding affinity for HSP90 as well as enhanced anti-tumor activity [169,170]. Paclitaxel, also named taxol, was another broadly used chemotherapeutic agent for various types of cancers. Paclitaxel showed a mitotic inhibitory effect via restricting the disassembly of microtubule polymers. The treatment of paclitaxel in cancer cells leads to defects in mitotic spindle assembly, chromosome segregation and cell division [229]. Further affinity purification experiments with biotinylated taxol also found the association of HSP90, although the binding region is unknown [171]. Sansalvamide A (San A) was a depsipeptide extracted from *Fusarium* that manifested moderate anti-tumor activity in cancer cells. Gu W. et al., developed San A-amide, a derivative of San A, which exhibited a better anti-proliferative efficacy than its natural form and inhibited the interaction between HSP90 and several co-chaperones such as IP6K2, FKBP52 and HOP [172,230,231]. Deguelin, which was first found in *Derris trifoliate* or *Mundulea sericea*, also showed the inhibitory activity in various cancer cell lines and pre-clinical models [232]. Biochemical research demonstrated that deguelin binds to the C-terminal ATP-binding pocket of HSP90 to inhibit the association with various client proteins such as Akt, IKK, NF-κB, mTOR, survivin and HIF-1α [174,233]. Considering that the high dose of deguelin was associated with Parkinson’s disease-like syndrome, Lee S. et al., developed a new deguelin derivative, L80, which showed comparable pro-apoptotic activity both in vitro and in vivo without obvious side effects [234].

Besides chemical inhibitors, it is also achievable to inhibit HSP90 by synthesizing peptidyl mimicry. In 2005, Plescia J. et al., identified the minimal survivin sequence of K79-L87 for its binding to the “shepherding” chaperone HSP90. It was named shepherdin, which exhibited the high binding affinity for the ATP pocket of HSP90, resulting in the degradation of its client proteins, especially the survivin. Interestingly, shepherdin inhibits the growth of HeLa cells by inducing mitochondrial apoptosis, with the release of mitochondrial cytochrome c in the cytosol. This indicates that there is a probability of the direct mitochondrial targeting of shepherdin [26,175].

### 4.2. Inhibitors of HSP90 Co-Chaperones and Clients

PU-WS13 was identified in a high-throughput compound screening, which selectively inhibited the mitochondria HSP90-GPR94. PU-WS13 possessed anti-proliferative activity by disrupting the interaction between GPR94 and HER2 at the cell membrane, leading to the lysosomal degradation of HER2 [235]. Celastrol was first extracted from the root of *Tripterygium wilfordii* as a traditional medicine to treat inflammatory and autoimmune diseases [236]. Several groups have shown that celastrol exhibits cytotoxicity against various cancer cell lines [237,238,239,240]. Celastrol is a quinone methide triterpene and covalently binds to the cysteine residues on CDC37 while sparing HSP90 [241], which results in the degradation of HSP90-dependent client kinases [242]. Gedunin was first isolated from the *Azadirachta indica*, which was widely used for treating malaria and other infectious diseases [243]. Gedunin not only effectively inhibits multiple cancer cell lines, but also activates heat shock response [243]. Mechanistically, Patwardhan C. et al., found that gedunin binds to co-chaperone p23 and disrupts its interaction with HSP90, which is detrimental for the HSP90 folding machinery [243]. Another HSP90-CDC37 disruptor, Withaferin A (WA), was first isolated from *Withania somnifera* and used for inflammatory diseases [244]. Similarly, it exerted an anti-proliferative property in numerous cancer cell lines, likely through inducing the degradation of multiple client proteins [244,245,246,247]. Derrubone was originally isolated from *Debrris robusta*, and its anti-tumor effect was also manifested in a number of cancer cell lines by stabilizing HSP90–client interactions and preventing the HSP90 chaperone cycle through its reaction cycle [248]. It was shown that derrubone disrupted the HSP90–CDC37 hetero-complex and resulted in the degradation of eIF2α kinase [249]. Gambogic acid was extracted from the *Garcinia hanburyi* and has already been used to treat infectious disease or cancer [250]. Since gambogic acid showed efficacious activity by controlling the proliferation, angiogenesis, and metastasis of multiple cancer cell lines, it was under evaluation at phase II clinical trials in China for patients with metastatic malignancies [251]. Cruentaren A was isolated from the *Byssovorax cruenta*, which exerts cytotoxicity in histiocytic lymphoma U-937 cells through selectively inhibiting mitochondrial ATP synthase, which may also be a co-chaperone of HSP90 [252,253].

### 4.3. Nanoparticles for HSP90 Inhibitors Delivery

In general, HSP90 inhibitors were delivered by the oral or intravenous approach. However, they were always accompanied by adverse effects during the treatment course. To avoid the cytotoxicity in the normal counterpart, spatio-temporally controlled nanoparticles containing HSP90 inhibitors emerged as a new path to be tested in preclinical models. In a study, Yang M. et al., used a versatile single-step surface-functionalizing technique to prepare a 17-AAG oral delivery system using PLGA/PLA-PEG-FA nanoparticles (NP-PEG-FA/17-AAG) to treat colitis-associated cancer (CAC). They showed enhanced efficacy in CAC therapy while reducing systemic exposure [254]. Another study engineered a nanoparticle (NP) containing both docetaxel and radicicol (DocRad-NP), wherein radicicol is conjugated to cholesterol and held in the lipid bilayer. The treatment of breast cancers with DocRad-NP optimally re-primes NK cells via the prolonged induction of NK-activating ligand receptors and the temporal control of drug release [255]. Other microcarrier nano-platforms also hold great potential to improve the standard care of therapy for patients with cancers. For example, 17-AAG could be efficiently encapsulated into nanoporphyrins (NP-AAG), which can generate efficient heat and ROS simultaneously with light activation at the tumor sites for dual-modal photothermal- and photodynamic-therapy (PTT/PDT) [256]. It has been reported that bovine serum albumin (BSA) nanoparticles (NPs) can also serve as carriers for anti-cancer drugs. For example, Rochani A. et al., showed that luminespib-loaded BSA NPs can be used in vitro for the investigation of cancer therapy in MIA PaCa-2 and MCF-7 cancer cells [257].

### 4.4. Mechanism of Resistance to HSP90 Inhibitors

Although HSP90 inhibitors exhibited considerable anti-tumor efficacy, the emergence of drug resistance limited their prolonged benefit. Elevated levels of HSPs and heat shock factor 1 (HSF1) are typically associated with drug resistance and poor clinical outcomes in various malignancies. Samarasinghe B. et al., showed that the activation of HSF1 confers resistance to HSP90 inhibitors (GA or 17-AAG) through up-regulating sequestosome 1 and promoting the autophagic flux [258]. HSF1 was also identified through a pooled RNA interference screen, and the combination of *HSF1* knockdown with HSP90 inhibitors exhibited a marked effect on various cancer cell lines and tumor mouse models [259]. In glioblastoma cells, low NQO1 activity is a potential mechanism of acquired resistance to 17-AAG, and such resistance can be overcome with another HSP90 inhibitor (VER-50589 or NVP-AUY922) [260]. Nevertheless, different mechanisms were found in different HSP90 inhibitors. Liu R. et al., found that SLC7A11 expression shows a negative correlation with the growth inhibitory potency of geldanamycin, but not with its analog 17-AAG. The ectopic expression of SLC7A11 in HepG2 cells confers resistance to geldanamycin, but not to 17-AAG, partly as a result of the differential dependence on ROS for cytotoxicity [261].

## 5. Discussion

Although there are dozens of HSP90 inhibitors that have entered clinical trials for treating a broad range of tumors (Figure 2) [262], only two inhibitors (panobinostat and irsogladine) that inhibit HSP90 as a secondary target have been approved by the FDA [214,215]. The main obstacle has been the intolerant adverse effects such as hepatotoxicity, fatigue, nausea, diarrhea, myalgias, and retinal dysfunction, leading to a plausible notion that HSP90 may not be a viable anti-cancer target [263,264]. Another impediment was that neither a predictive nor pharmacodynamic marker was developed to select a subset of patients who might potentially benefit [262]. Thus, personalized therapies may lead to improved efficacy since a subset of patients with tumors that over-express HSP90 could be selected. Therefore, the sensitivity could be enhanced while the cytotoxicity is minimized [20].

Considering that most side effects can be mitigated via a lower dosage of drug administration, we thus propose that rational-based combination therapies involving HSP90 inhibitors may be more efficacious for cancer treatments that target oncogenic signaling pathways in parallel with reduced doses (Table 3). For example, the combined low-dose treatment of rhabdomyosarcoma with the proteasome inhibitor Bortezomib (5–7.5 nM) plus the HSP90 inhibitor 17-DMAG (≤50 nM) exhibited greater efficacy than either single agent with improved side effects [265]. Another study found that the combination of a lower dose of docetaxel (5 mg/kg, three times a week) with IPI-504 (50 mg/kg, twice a week) showed enhanced antitumoral effects in multiple NSCLC xenograft models [266]. The combination therapeutics with more appropriate drug dosage in the clinic warrants further investigation.

A significant amount of previous work has illustrated that the quantity of the subcellular localization of HSP90 is quite different between tumor and normal tissues, especially in mitochondria and the extracellular matrix [267,268]. Further, the high expression levels of mitochondria and extracellular HSP90 have been associated with various types of cancers, with very low levels detected in the normal counterpart. Therefore, inhibitors can be rationally designed, specifically for targeting mtHSP90 or eHSP90 [44,48,267]. Gamitrinibs are mtHSP90 inhibitors that showed substantial anti-tumor activity with minimal to no side effects, and they have been under clinical evaluation [139,191]. While the selective inhibitors of eHSP90 are still in development, the concept of developing organelle-specific drugs is being tested not only for HSP90 inhibitors, but also for other therapeutics targeting organelle-specific oncoproteins, with the ultimate goal of improving drug activity and minimizing side effects [96].

## Figures and Tables

**Figure 1 cells-11-02778-f001:**
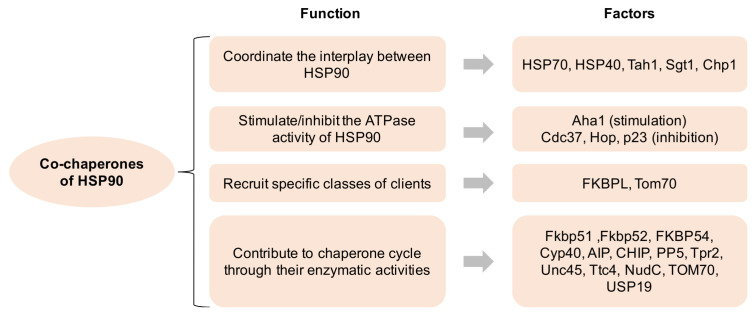
Co-chaperones of HSP90 and its function. HSP90 comprises four isoforms: (1) HSP90α (an inducible form) and HSP90β (a constitutive form) are mainly located in the cellular cytosol; (2) the glucose-regulated protein (GRP94) is localized in the endoplasmic reticulum; (3) Hsp75/tumor necrosis factor receptor associated protein 1 (TRAP-1, also known as mitochondrial HSP90) is located on the mitochondria [7,8]. All HSP90 isoforms play critical roles in cancer, neurodegenerative disorders, and other disease states, indicating that their pharmacological targeting may have profound implications for the treatment of these illnesses [9].

**Figure 2 cells-11-02778-f002:**
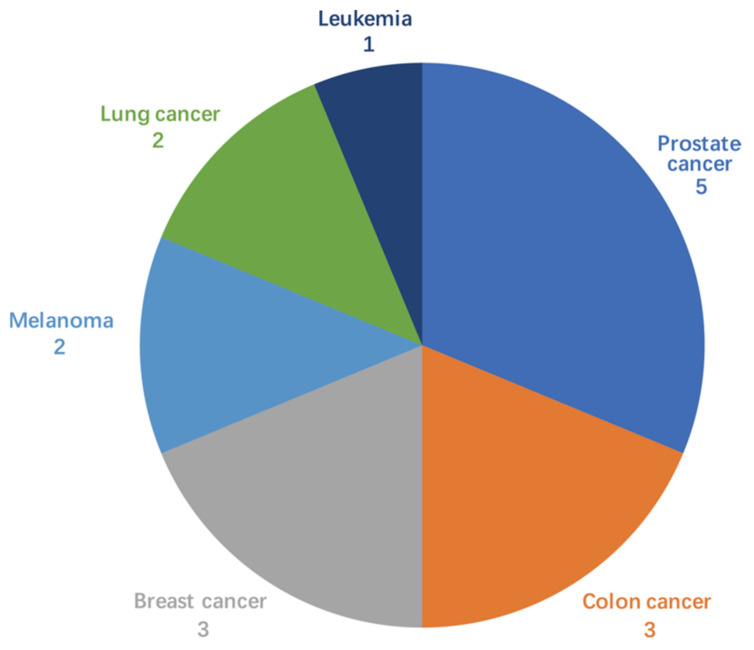
Number of HSP90 inhibitors used in different cancer types under clinical evaluation (only recruited/active/completed were counted).

**Table 1 cells-11-02778-t001:** HSP90 clients are associated with hallmarks of cancer.

Phenotype	Clients	References
Uncontrolled proliferation	EGFR, HER2, RAF1, CDK4, Akt, BCR-ABL, v-Src, c-Src, FAK, CKII, CHK1, eIF-2α kinase	[6,16,17,54,55,56,57,58,59,60,61,62]
Anti-apoptosis	p53, Akt, Survivin, IKK, NF-κB, PLK, WEE1, Myc, CDK4, CDK6	[6,42,55,63,64,65,66,67,68]
Angiogenesis	HIF1α, Akt, EGFR, HRE2, FLT3, VEGFR2	[19,69,70,71,72,73]
Immortalization	Telomerase	[18]
Invasion/Metastasis	MMP2, MMP9, c-MET	[19,34,74,75]
Others	Glucocorticoid receptor, Mineralocorticoid receptor, Progesterone receptor, Estrogen receptor, Androgen receptor, Oestrogen receptor, Nitric oxide synthase, Centrin/centrosome, Calmodulin, MDM2, UHRF1, BRCA2, OCT4, Nanog, STAT3, Calcineurin, CFTR, NLR proteins, RAD51/RAD52, Tau, HCK, JAK1 and/or JAK2	[76,77,78,79,80,81,82,83,84,85,86,87,88,89,90,91,92,93,94]

**Table 2 cells-11-02778-t002:** HSP90 inhibitors.

Type	Order	Inhibitor	Alias/Description	Structure	HSP90 Binding Site	Organelle selectivity	Manufacturer	Application	Drug Used for Combination	Recruting or Active Clinical Stage	Reference
GA and its derivatives	1	Geldanamycin	GA, NSC 122750	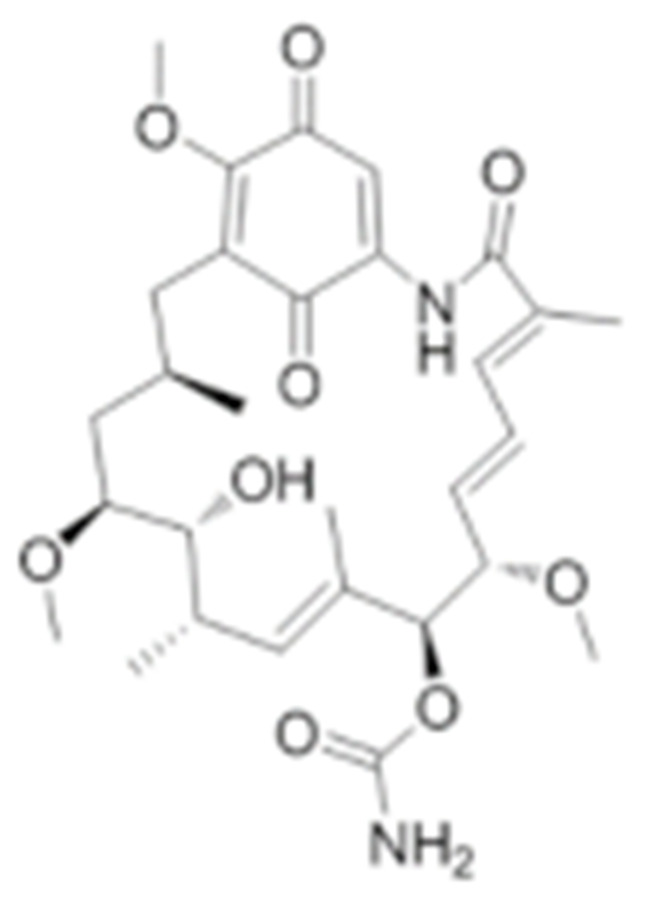	N-terminal ATP-binding pocket		NCI	melanoma, leukemia, colorectal cancer, prostate cancer, lung cancer, breast cancer, kidney cancer, bladder cancer, gastric cancer, head and neck cancer, ovarian cancer, neuroblastoma, osteosarcoma	docetaxel, irinotecan hydrochloride,	Phase I/II	[126,127]
	2	17- AAG	Tanespimycin	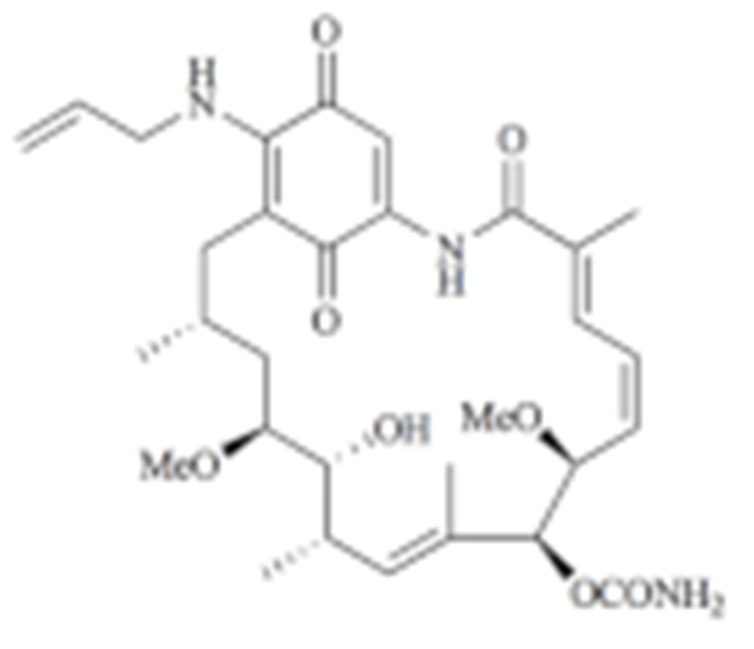	N-terminal ATP-binding pocket	Cytosol	Pfizer	thyroid cancer, lymphoma, leukemia, prostate cancer, neuroblastoma, osteosarcoma, sarcoma, lung cancer, myeloma, kidney cancer, ovarian epithelial cancer, pancreatic cancer, breast cancer	irinotecan hydrochloride, sorafenib tosylate, cytarabine, docetaxel, gemcitabine hydrochloride, everolimus, Bortezomib	Phase I/II/ III	[128,129]
	3	17- DMAG	Alvespimycin	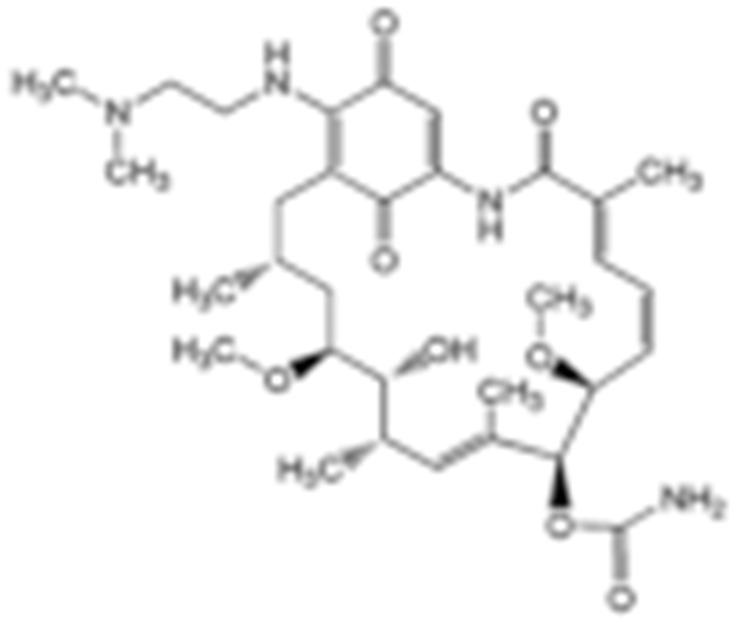	N-terminal ATP-binding pocket		NCI	lymphoma, breast cancer, lung cancer, gastric cancer, prostate cancer, myeloma	Trastuzumab	Phase I	[130,131,132]
	4	IPI-504	Retaspimycin	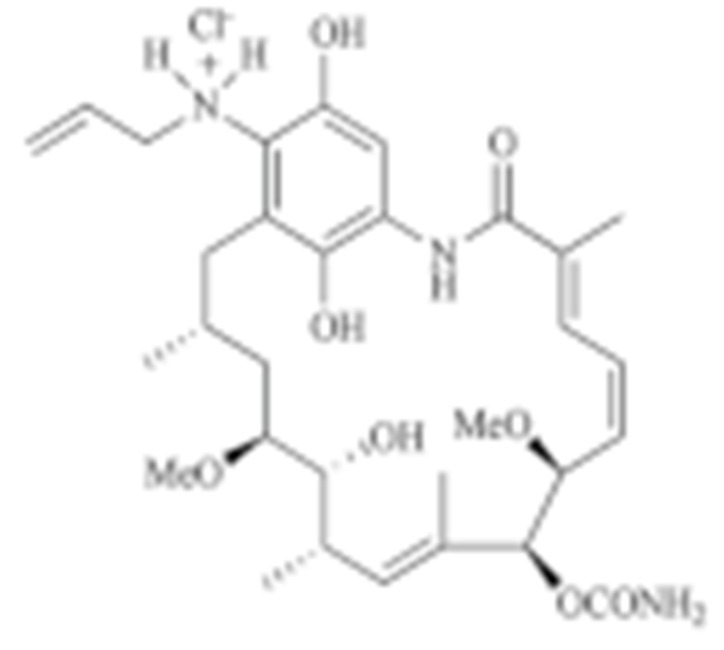	N-terminal ATP-binding pocket	Cytosol	Infinity	lung cancer, prostate cancer, myeloma, sarcoma, leukemia, lymphoma, pancreatic cancer, ovarian epithelial cancer, breast cancer, kidney cancer, bladder cancer, gastric cancer	docetaxel, everolimus, cytarabine, gemcitabine hydrochloride, irinotecan hydrochloride, sorafenib tosylate, Bortezomib, imatinib mesylate	Phase I/II/ III	[133]
	5	NVP-BEP800	VER82576	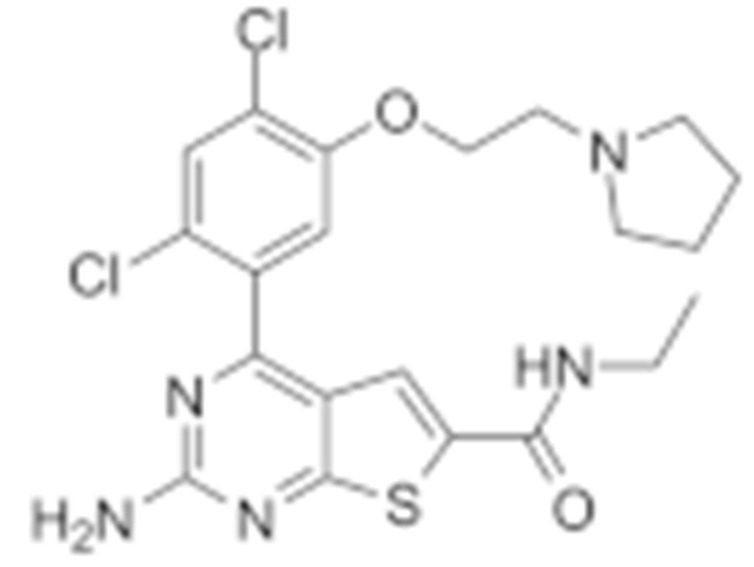	N-terminal ATP-binding pocket						[134,135]
	6	TAS-116	Pimitespib	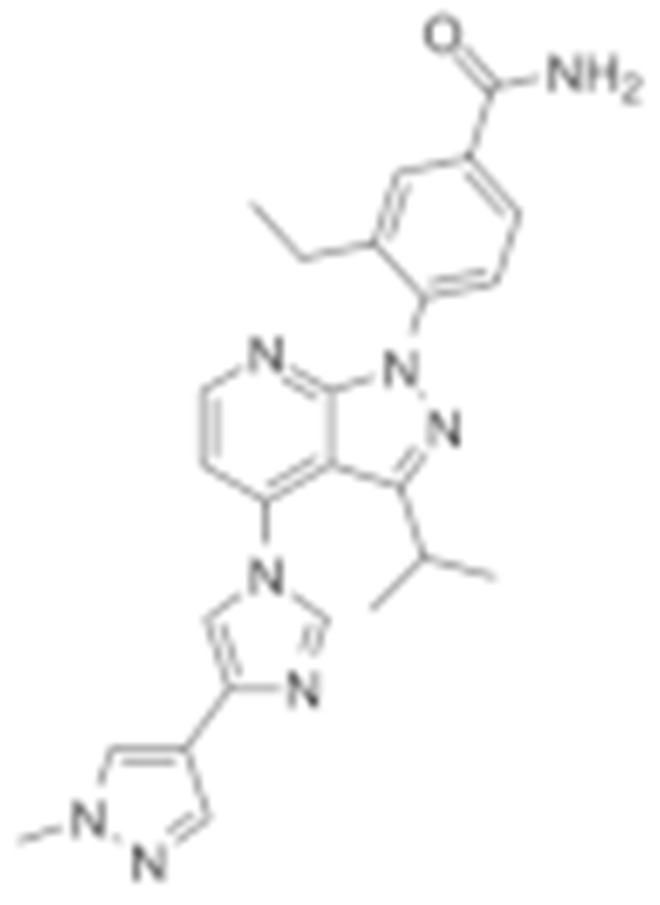	N-terminal ATP-binding pocket	Cytosol	Taiho	Gastrointestinal cancer, pancreatic cancer, lung cancer, colorectal cancer		Phase I	[136,137]
	7	Gamitrinib TPP		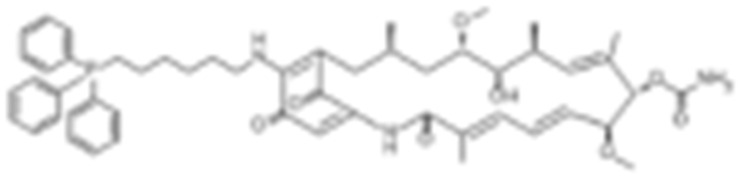	N-terminal ATP-binding pocket	Mitochondrion		lymphoma		Phase I	[138,139]
	8	SNX-2112	PF-04928473	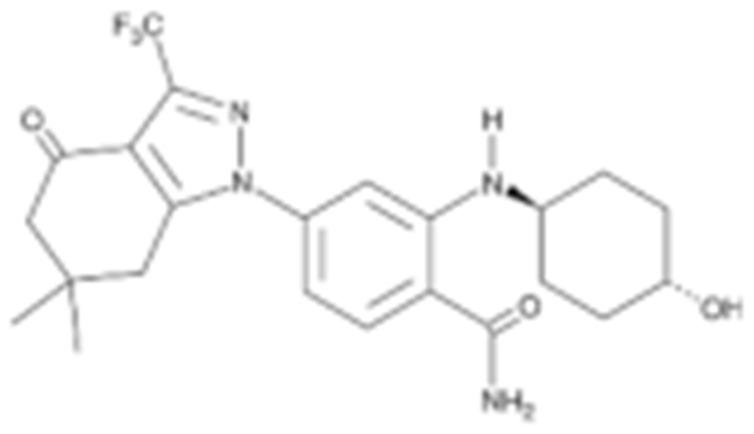	N-terminal ATP-binding pocket		Serenex				[140]
	9	Macbecin		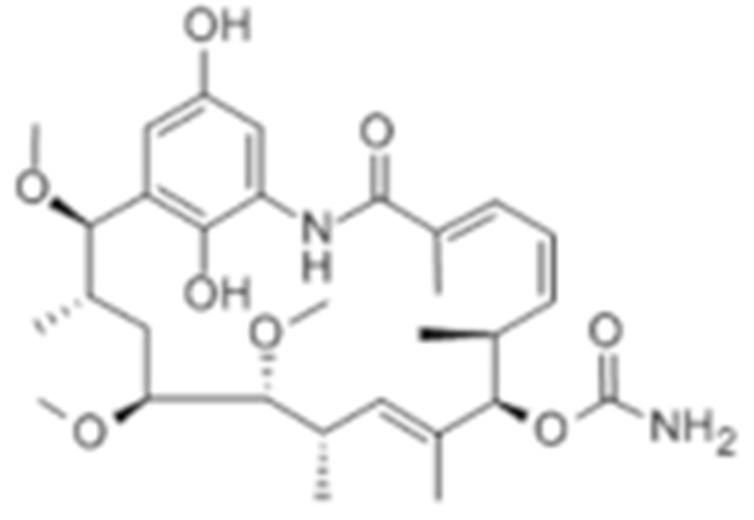	N-terminal ATP-binding pocket						[141]
	10	XL888		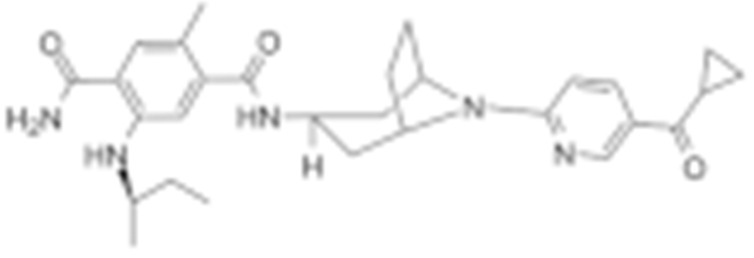	N-terminal ATP-binding pocket		Exelixis	colorectal cancer, myeloma, pancreatic cancer	Vemurafenib, Cobimetinib	Phase I	[142,143,144]
Radicicol or resorcinol-containing derivatives	11	Radicicol	RA, RDC	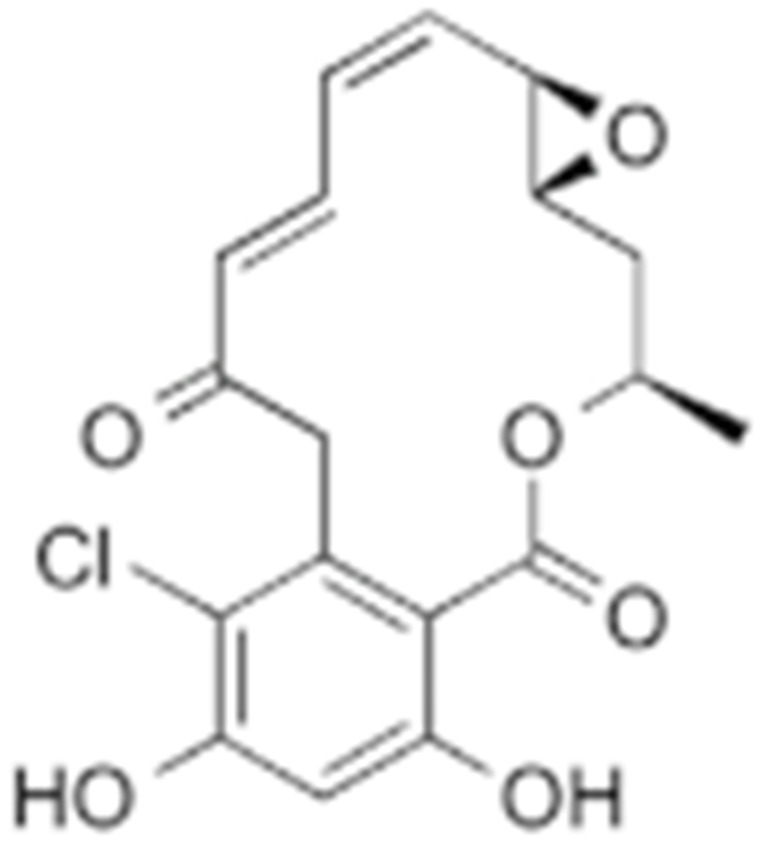	N-terminal ATP-binding pocket						[127,145,146]
	12	STA-9090	Ganetespib	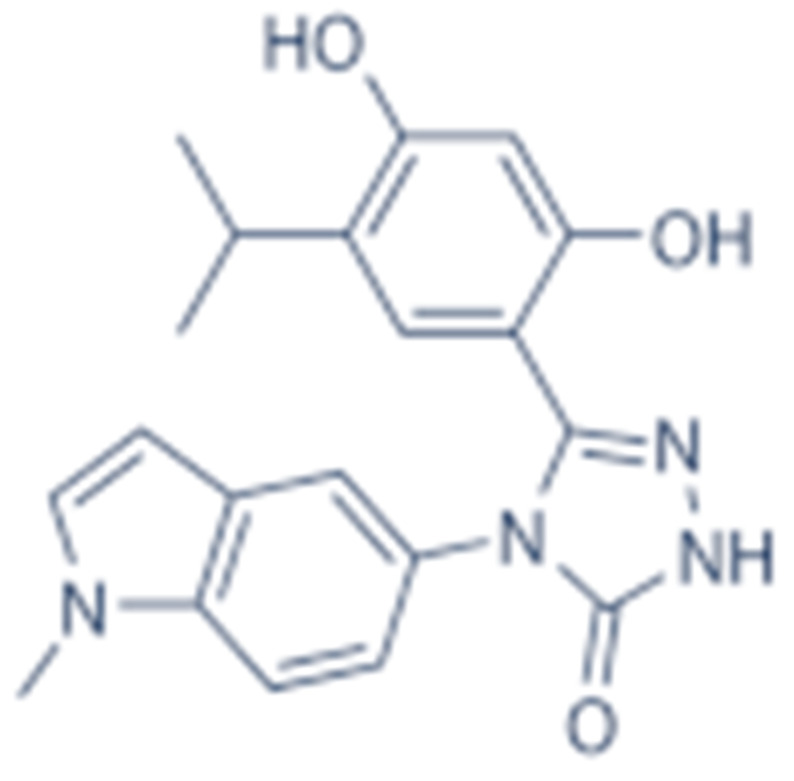	N-terminal ATP-binding pocket		Synta Pharmaceuticals	Hepatocellular Carcinoma, Esophagogastric Cancer, melanoma, breast cacncer, lung cancer, colorectal cancer, prostate cancer, leukemia, myeloma, ovarian cancer	Docetaxel, crizotinib, Sirolimus, capecitabine, Bortezomib, Dexamethasone, Fulvestrant, Carboplatin	Phase I/II	[147,148]
	13	CCT018159		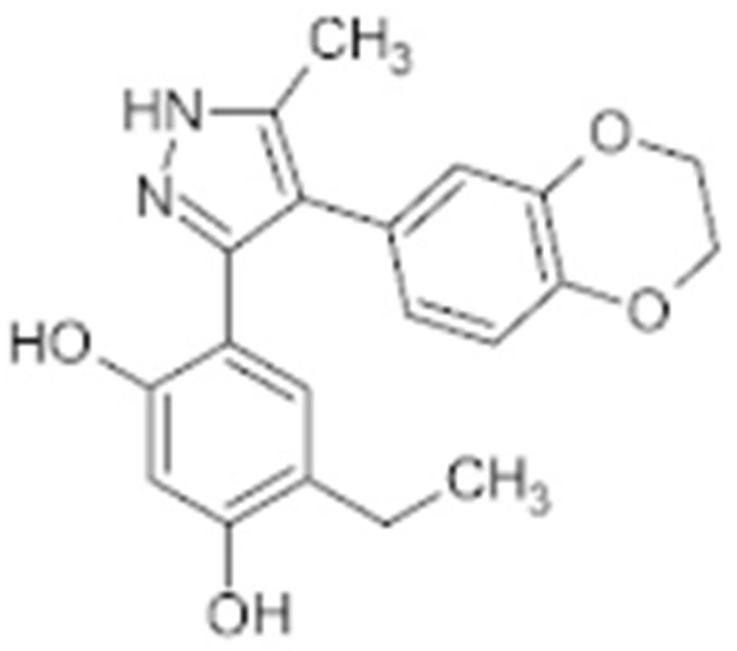	N-terminal ATP-binding pocket						[149,150]
	14	NVP-AUY992	Isoxazole/luminespib	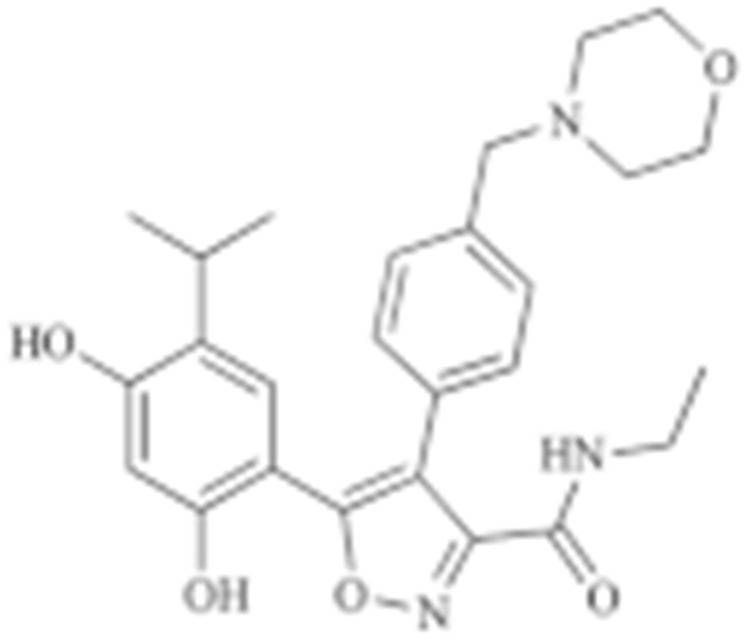	N-terminal ATP-binding pocket		Novartis				[151]
	15	KW-2478		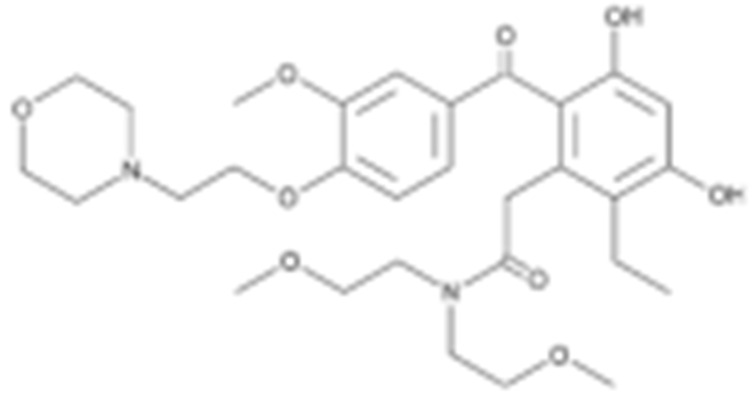	unknown		Kyowa Hakko Kirin		Bortezomib	Phase I/II	[152]
	16	AT13387		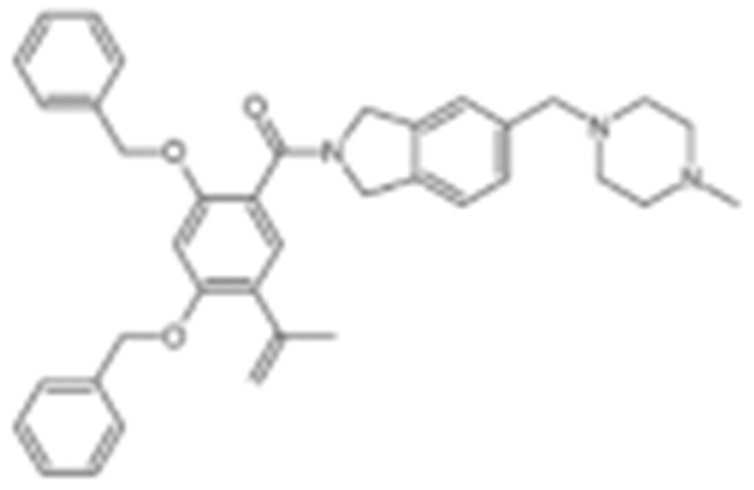	N-terminal ATP-binding pocket		Astex Pharmaceuticals	Gastrointestinal Stromal Tumors, pancreatic cancer, lung cancer, breast cancer	Crizotinib, Imatinib, abiraterone acetate, Prednisone, Onalespib	Phase I/II	[153,154]
GA and RA chimeric molecules	17	Radanamycin		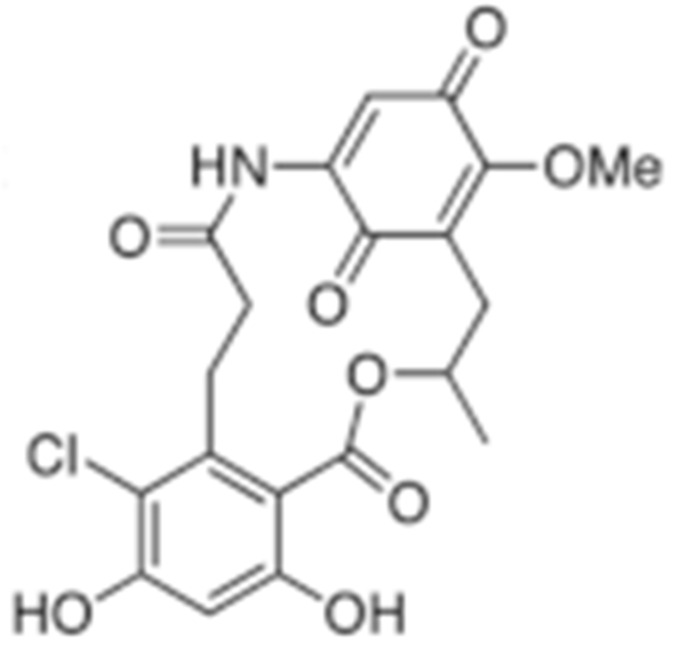	N-terminal ATP-binding pocket						[155]
	18	Radamide		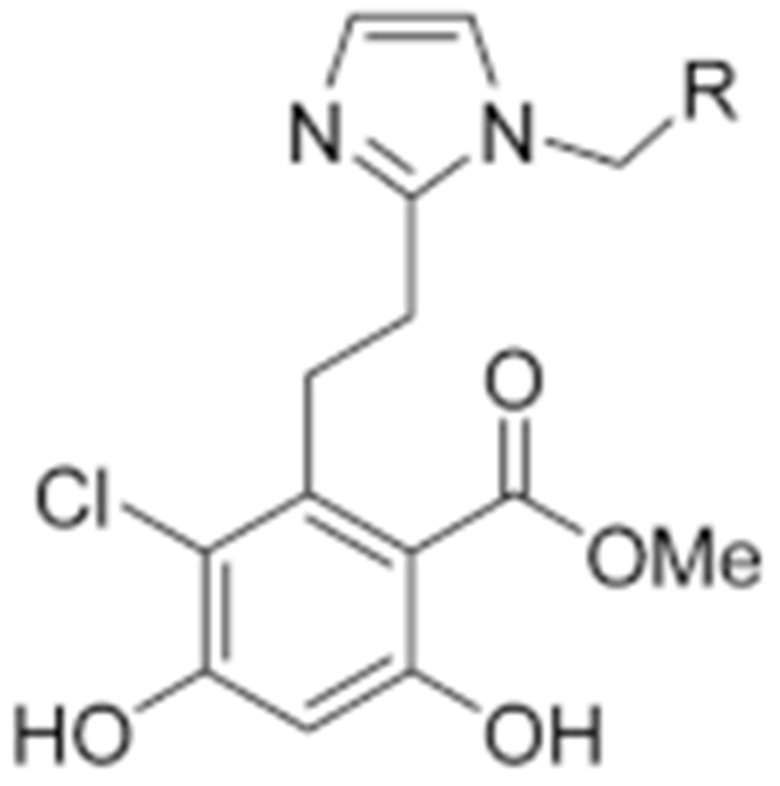	N-terminal ATP-binding pocket						[155]
	19	Radester		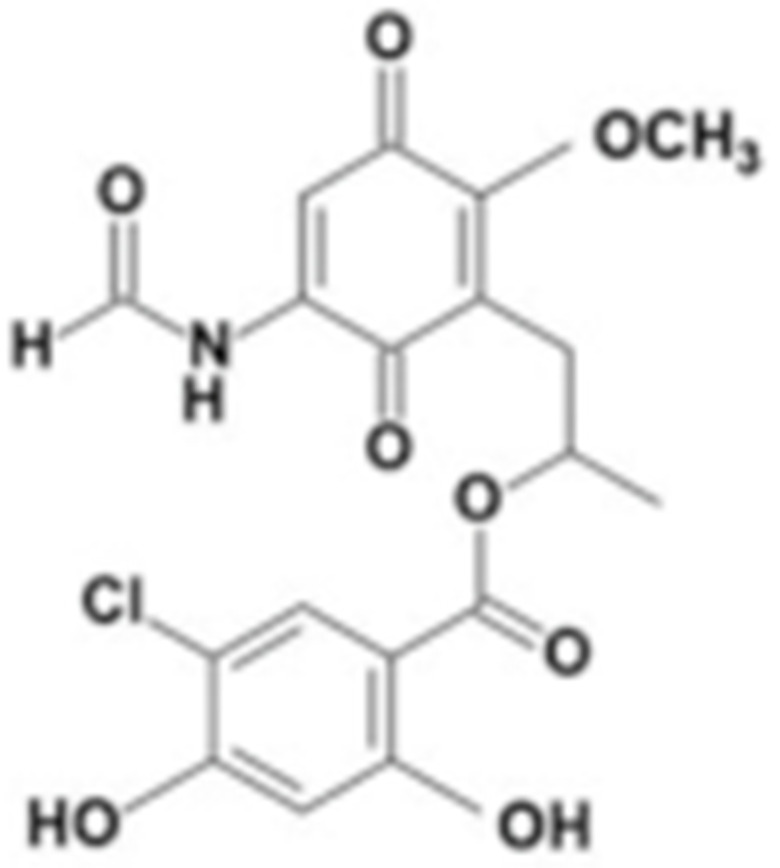	N-terminal ATP-binding pocket						[155]
Purine-based molecules	20	PU3		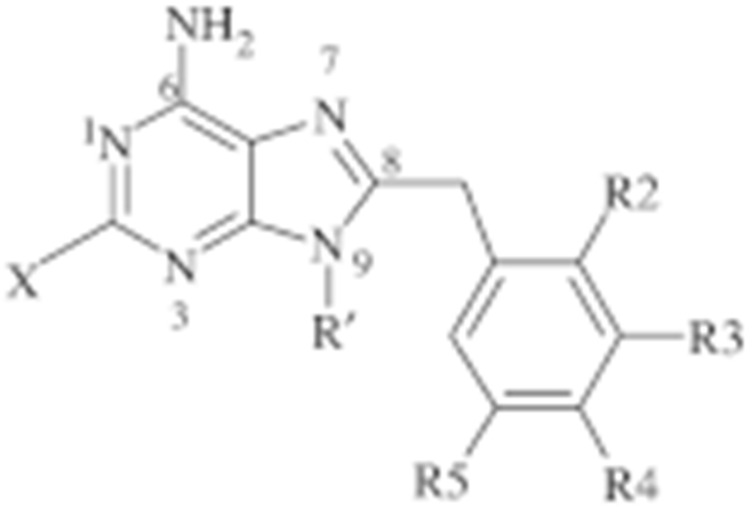	N-terminal ATP-binding pocket					Phase I	[156]
	21	BIIB021	CNF2024	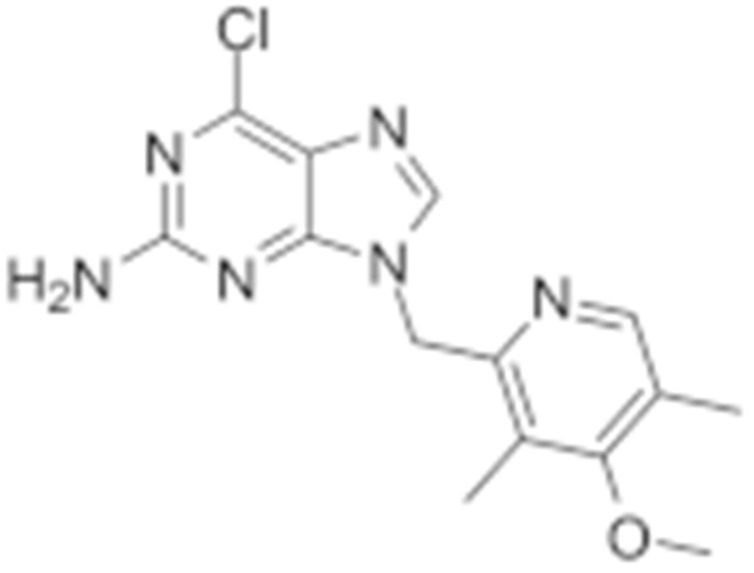	N-terminal ATP-binding pocket		Biogen Idec	breast cancer, lymphoma	exemestane (Aromasin), trastuzumab	Phase I/II	[157,158]
	22	BIIB028		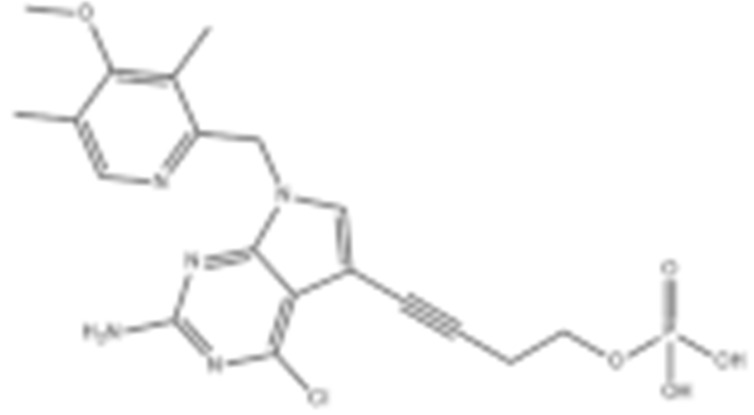	N-terminal ATP-binding pocket		Conforma Therapeutics	breast cancer, melanoma, gastrointestinal cancer, lymphoma, myeloma	Onalespib, Bortezomib, Cetuximab	Phase I/II/ III	[159]
	23	DN401		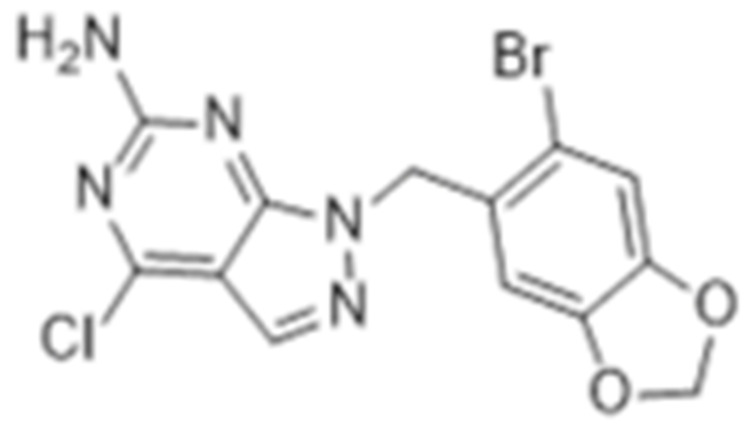	N-terminal ATP-binding pocket						[9,160]
	24	MPC-3100		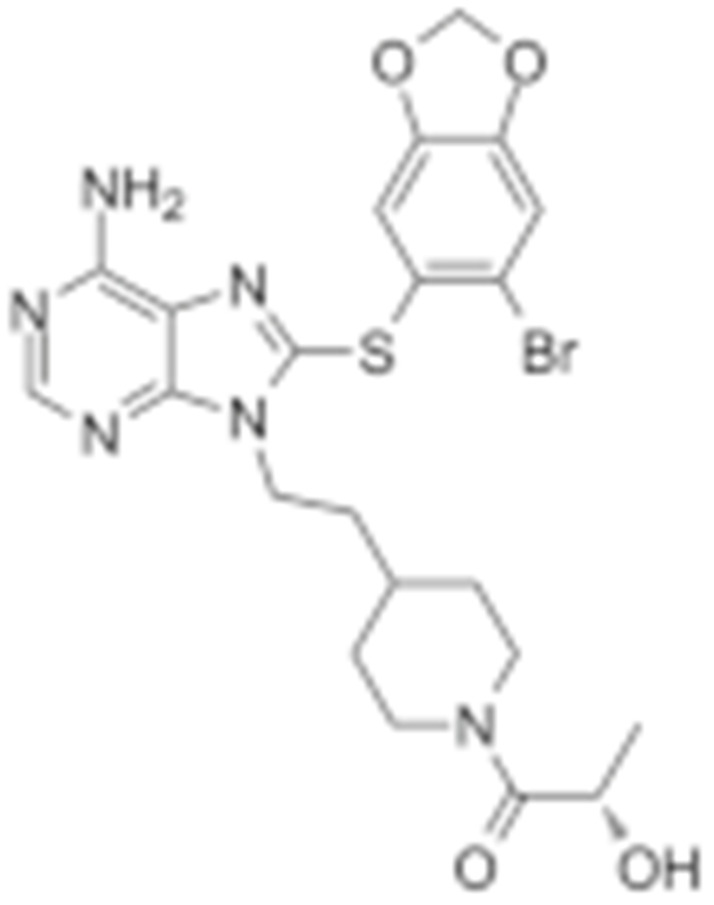	N-terminal ATP-binding pocket		Myriad Pharmaceuticals			Phase I	[161]
Other inhibitors	25	Panobinostat	Farydak	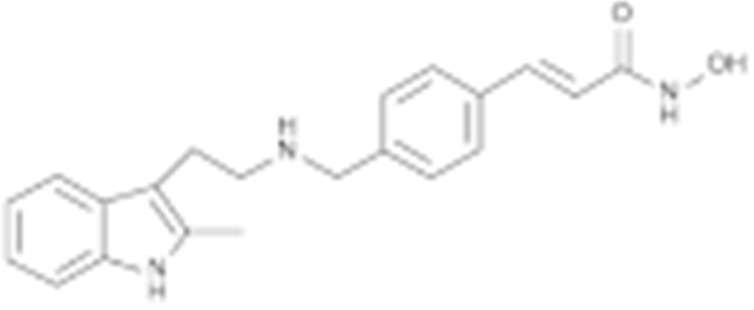			Novartis			Phase I/II/ III	
	26	Novobiocin	Albamycin/ cathomycin	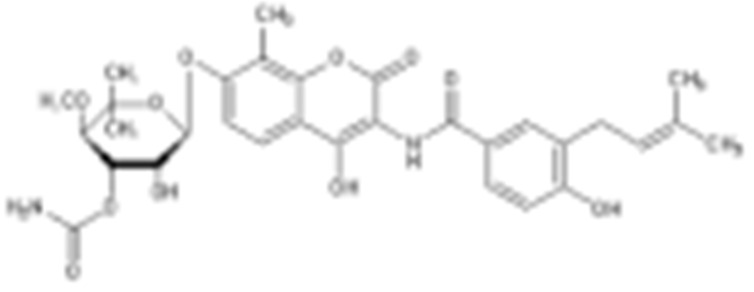	C-terminus		Pharmacia & Upjohn				[162]
	27	EGCG	Epigallocatechin gallate	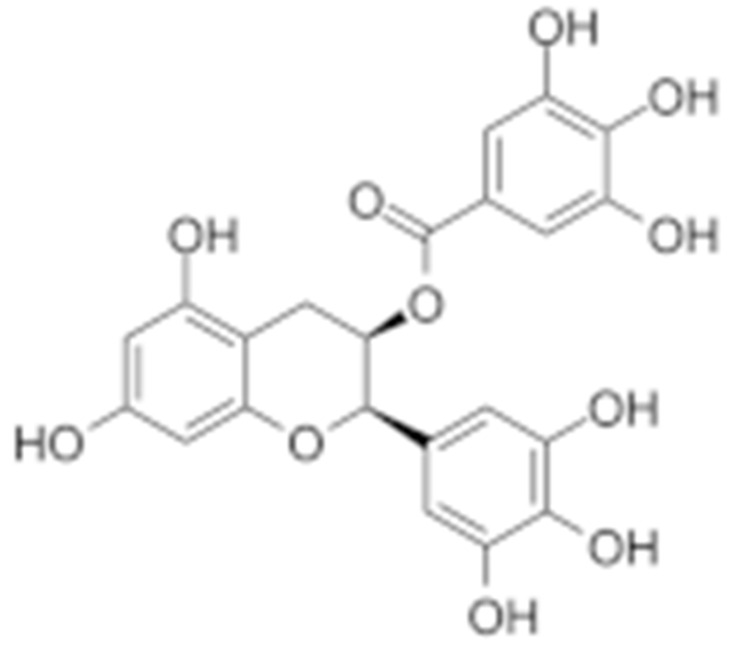	C-terminus			colon cancer, prostate cancer, bladder cancer, head and neck cancer, breast cancer, lung cancer, pancreatic cancer	Sunphenon, Erlotinib, Polyphenon E	Phase I/II/ III/IV	[163,164]
	28	Silybin		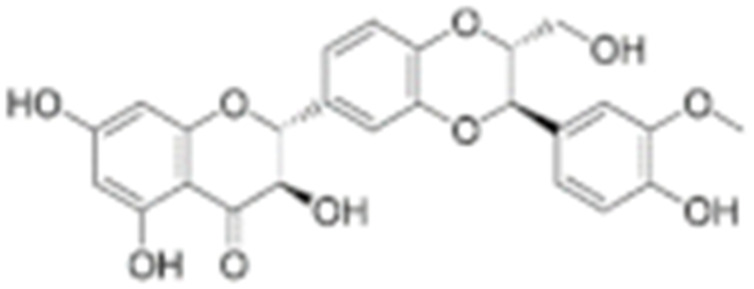	C-terminus			prostate cancer		Phase I/II/ III/IV	[165]
	29	Cisplatin		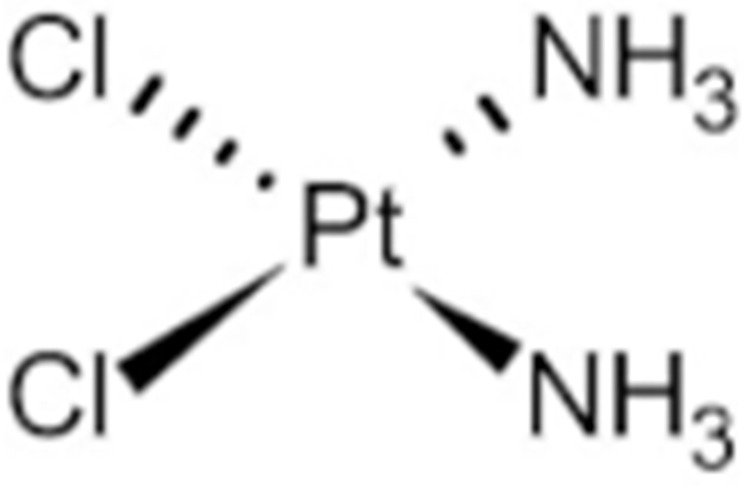	C-terminus					Phase I/II/ III/IV	[166,167,168]
	30	LA-12	Cisplatin derivative	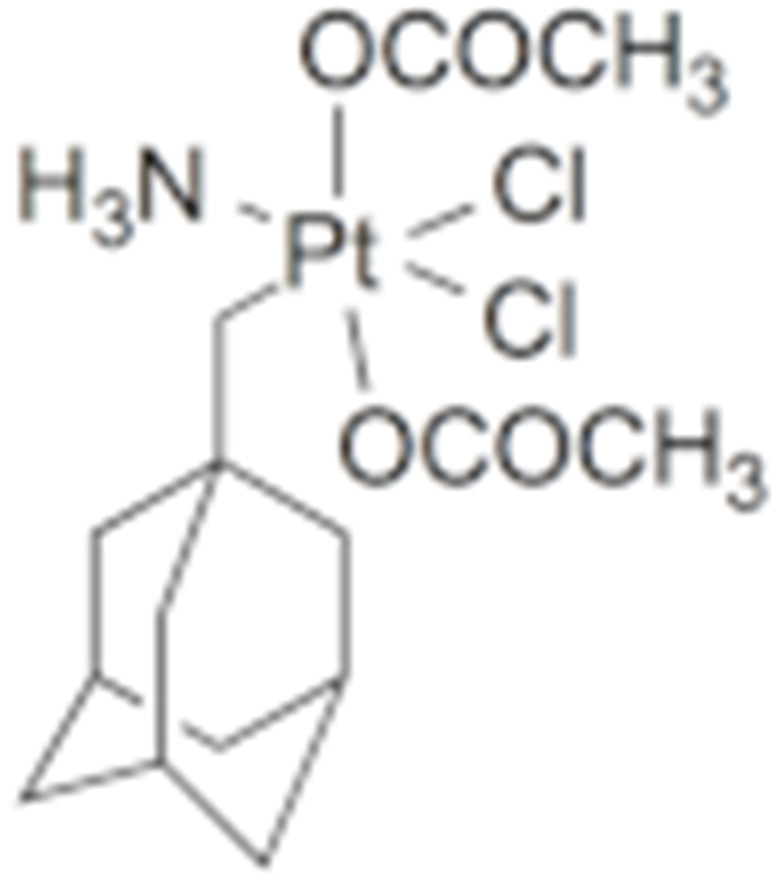	C-terminus					Phase I/II/ III/IV	[168,169,170]
	31	Paclitaxel	Taxol	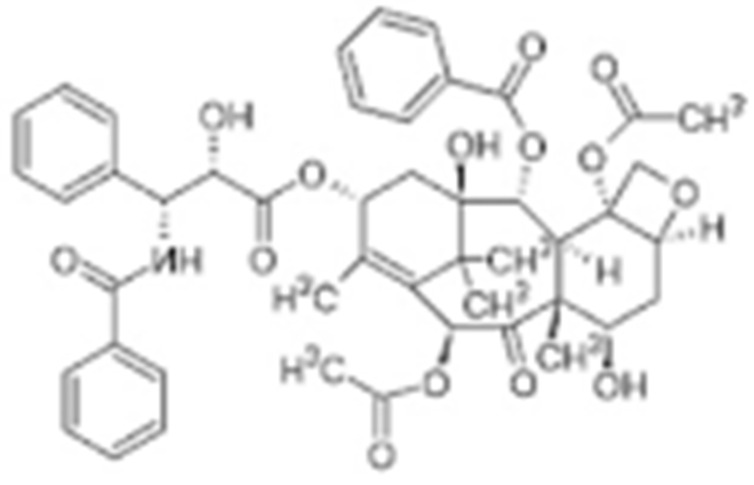	unkonwn					Phase I/II/ III/IV	[171]
	32	Sansalvamide A	San A	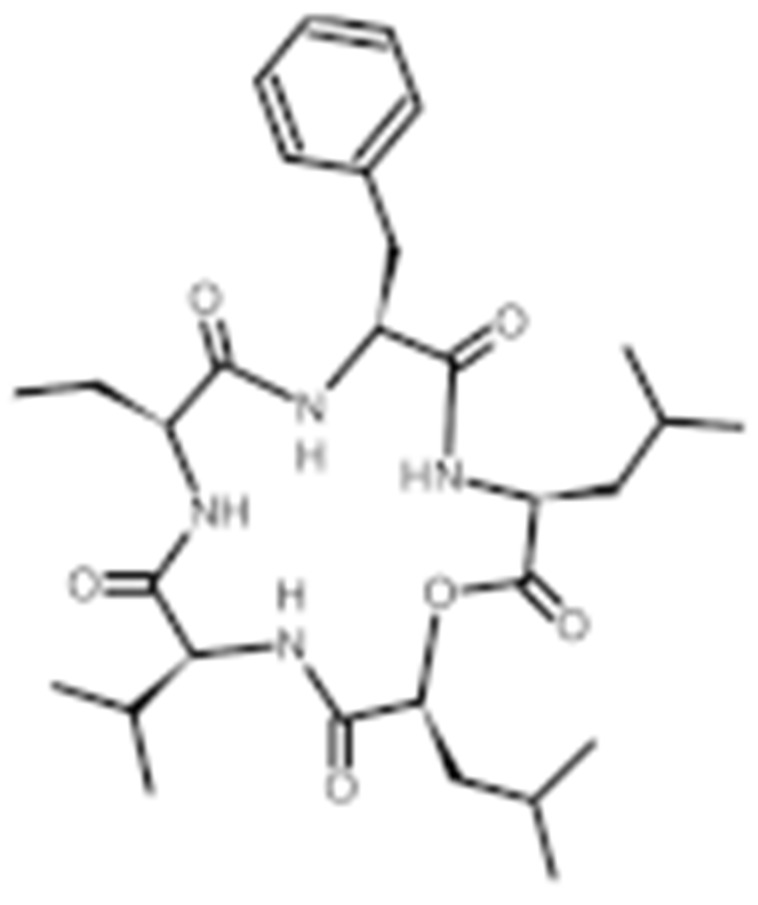	N-MD						[172,173]
	33	L80	Deguelin derivative	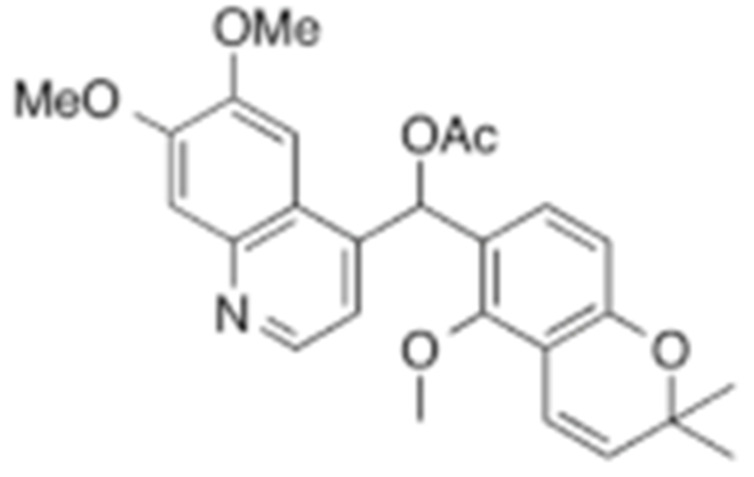	C-terminal ATP-binding pocket					Phase III	[174]
	34	Shepherdin	/	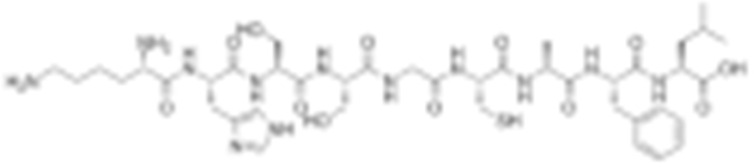	N-terminal ATP-binding pocket	Mitochondrion					[26,175]
	35	SNX-5422	PF-04929113	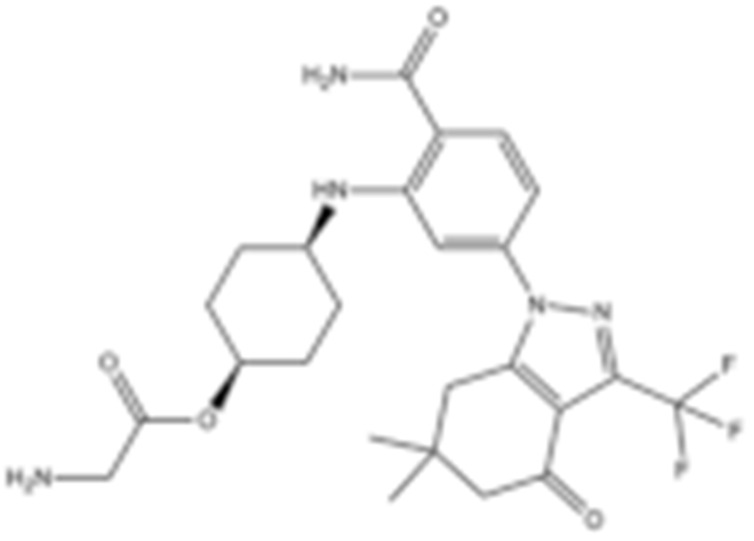	N-terminal ATP-binding pocket		Pfizer	leukemia, lymphoma.		Phase I	[176,177]
	36	HSP990	NVP- HSP990	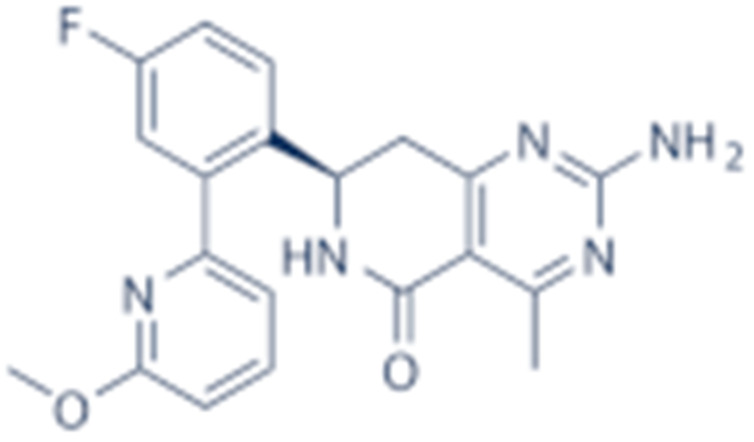	N-terminal ATP-binding pocket		Novartis			Phase I	[178]
	37	Pseudolaric Acid A	PAA	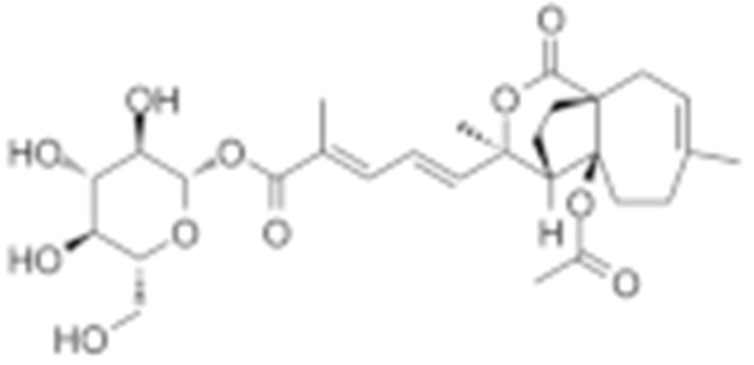	N-terminal ATP-binding pocket						[179]
	38	CH5138303		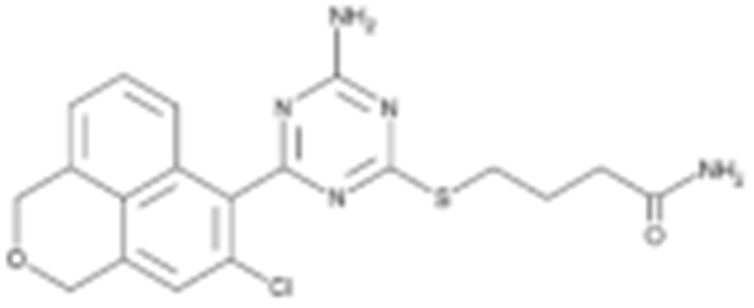	N-terminal ATP-binding pocket						[180]
	39	NMS-E973	Isoxazole derivative	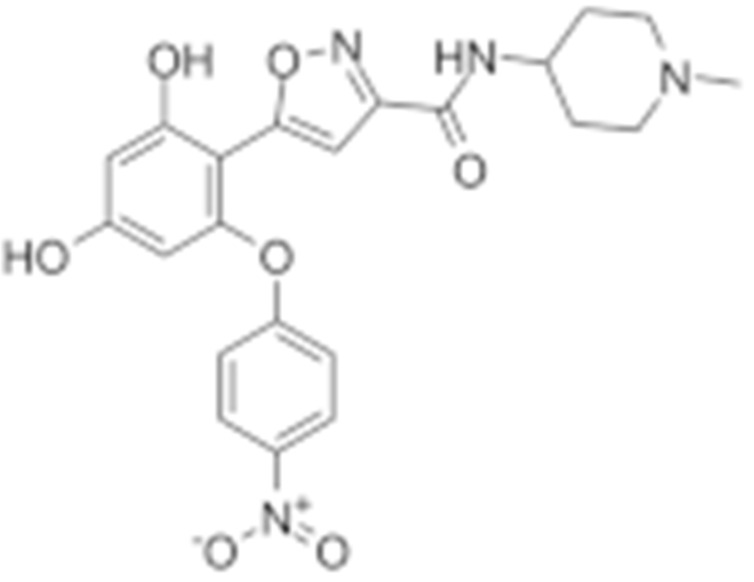	N-terminal ATP-binding pocket		Nerviano Medical Sciences S.r.l. laboratories				[181]

**Table 3 cells-11-02778-t003:** FDA-approved drugs used for combination with HSP90 inhibitors in clinical trials *.

Drug	Number of Clinical Trials
Bortezomib	5
Docetaxel	4
Irinotecan	3
Sorafenib	2
Cytarabine	2
Gemcitabine	2
Everolimus	2
Trastuzumab	2
Imatinib	2
Crizotinib	2
Onalespib	2
Vemurafenib	1
Cobimetinib	1
Sirolimus	1
Capecitabine	1
Dexamethasone	1
Fulvestrant	1
Carboplatin	1
Abiraterone	1
Prednisone	1
Exemestane (Aromasin)	1
Cetuximab	1
Sunphenon	1
Erlotinib	1
Polyphenon E	1

* Only recruited/active/completed clinical trials were counted.

## Data Availability

Not applicable.

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
