# Peer review of "Targeting HSP90 as a Novel Therapy for Cancer: Mechanistic Insights and Translational Relevance"

_cells, 2022, doi:10.3390/cells11182778_

Round 1

Reviewer 1 Report

1) This review did not learn anything from reading this manuscript. A review article must have its own distinct focus. Instead, this manuscript basically extracted the writings from the following reviews of others: 1) Trepel, J.; Mollapour, M.; Giaccone, G.; Neckers, L. Targeting the dynamic HSP90 complex in cancer. Nat. Rev. Cancer 2010, 10, 537-549; 2) Jhaveri, K.; Taldone, T.; Modi, S.; Chiosis, G. Advances in the clinical development of heat shock protein 90 (Hsp90) inhibitors in cancers. Biochim. Biophys. Acta, Mol 2012, 1823, 742-55; 3) Neckers, L.; Blagg, B.; Haystead.; T, Trepel, JB.; Whitesell, L.; Picard, D. Methods to validate Hsp90 inhibitor specificity, to identify off-target effects, and to rethink approaches for further clinical development. Cell Stress Chaperones 2018, 23, 467-82; 4) Sanchez, J.; Carter, T.R.; Cohen, M.S.; Blagg, B.S. Old and new approaches to target the Hsp90 chaperone. Curr. Cancer Drug Targets 2020, 20, 253-70; 5) Csermely, P.; Tamás, S.; Csaba, S.; Zoltán, P.; Gábor, N. The 90-kDa molecular chaperone family: structure, function, and clinical applications. A comprehensive review. Pharmacology & therapeutics 1998, 79, 129-168.

2) Lack of any analytical thoughts of important issues, instead, just of a laundry list of things copied and pasted from other reviewers. 

3) If there are only two or three sentences for each "Section", why not combine them all. What was the point to put down those two meaningless sentences?

4) "1%", "up to 5%".....the authors ned to verify where it came from ; cite original papers , not reviews! In fact there had never been any REAL done to support the "1%" and "5%" claims. You ought not cite rumors! 

Reviewer 2 Report

The authors tried to describe the role of HSP90 and significance of HSP90 inhibitors. However, this reviewer felt that this manuscript need some attracting comprehensive schemas to understand easily. Moreover, Table 2 should be improved.   Major comments;  1. page 1, line 32, The definiton of "Heat shock family" should be described. Heat shock family includes HSF family and the family of heat shock proteins. 2. Page 5, line 191, this reviewer recommend to present a schema illustrating the HSP90 and its client protein in cancer phenotype to understand easily for the readers. 3. Table 2 is not organized and therefore, iti is very difficult to understand. Each culumn should be improved by adjusting size and culumn spaces to recognize more easily. For examle, it is suggested to use abbreviations. 4. Page 13, section 4.1, I recommend to present the schema for the mechanism and their characteristics of each HSP90 inhibitors. 5. Page 2, line 46 and page 17, line 431, I sugges to present the chema illustrating the interaction of HSP90 and co-chaperone to understand their role for cancer survival and usefulness for HSP90 inhibitors.

Round 2

Reviewer 2 Report

The authors improved their manuscript significantly, however, I recognized some points to be checked. After revising them, it can be published.

1. I found that Fig. 2 in not well organized on my printed version. The authors should check if it is OK.

2. The title of Table 3 says that Statistical analysis of drug used, however, no statistical analysis was shown or described. Moreover, line 577, the authors described that "with reduced doses", but we can not know which combination is used in combination with the reduced dose of Hsp90 inhibitor. Therefore, the authors should improve the Table 3 or describe about it in the manuscript.
